

# floodX: Urban flash flood experiments monitored with conventional and alternative sensors

Matthew Moy de Vitry[1,2], Simon Dicht[1], João Paulo Leitão[1]

[1]Eawag: Swiss Federal Institute of Aquatic Science and Technology, 8600 Dübendorf, Switzerland
[2]Institute of Civil, Environmental and Geomatic Engineering, ETH Zürich, 8093 Zurich, Switzerland

*Correspondence to*: Matthew Moy de Vitry (matthew.moydevitry@eawag.ch)

**Abstract**

The datasets described in this paper are intended to provide a basis on which new methods for monitoring and modelling urban pluvial flash floods can be developed. Pluvial flash floods are a growing hazard to property and inhabitants' well-being in urban

areas. However, the lack of appropriate data collection methods is often cited as an impediment for reliable flood modelling, thereby hindering the improvement of flood risk mapping and early warning systems. In the floodX project, 37 controlled urban flash floods were generated and monitored in a flood response training facility with state-of-the-art conventional sensors in the drainage network, as well as alternative sensors on the surface, namely temperature probes and surveillance cameras. With these data, the technical feasibility of utilizing citizen science and computer vision for urban flood monitoring can be explored. The

floodX project stands out as the largest documented flood experiment of its kind, providing both conventional and alternative data types in parallel and at high temporal resolution. Besides describing the flash flood experiments and the resulting datasets, weaknesses in the data and lessons learned are also described. The main data package is openly available at http://doi.org/10.5281/zenodo.236878.

## 1    Introduction

### 1.1    The need for comprehensive urban flood data

Urban pluvial floods, which occur when precipitation cannot be fully assimilated by a city's drainage system, can cause substantial economic disruption and claim lives every year. In Switzerland, the damages caused by surface water runoff to urban infrastructure is around CHF 45 Million per annum[1], and the figure is proportionally similar in the United Kingdom (Evans, 2004). Numerical modelling is an essential tool for urban flood risk assessment and flood risk management. It allows flood

events to be simulated so that drainage system weaknesses can be identified and possible corrective solutions be evaluated. Currently, urban flash flood models are most often calibrated with monitoring data from sensors installed in the underground drainage network, if at all. This information, while sufficient for normal dry and wet weather conditions, does not directly inform on the situation above ground during flood events. The lack of surface flood information leaves model parameters such as ground roughness with large uncertainty (Hunter et al., 2008). The lack of surface flood data is regularly brought up in urban flood

modelling research (Fewtrell et al., 2011; Hénonin et al., 2015; Sampson et al., 2012; Schmitt et al., 2004) and is detrimental to the objective evaluation of flood models (Dottori and Todini, 2013). Additionally, the lack of data limits the detail in which

---

[1] Private Communication of the *Etablissement Cantonal d'Assurance*, Switzerland



events can be modelled (Ciervo et al., 2015). To further illustrate the need for urban flood calibration data, Leandro et al. (2011) proposed a method for circumventing the issue by using detailed physical models to generate virtual overland flow data.

Because of the unfavourable conditions encountered at the street level, contactless measurement methods have an advantage when compared to conventional sensors. The use of visual methods such as large scale particle image velocimetry (LSPIV) is now gaining momentum for measuring flood events, as researchers start leveraging social media and crowdsourcing to collect data (Le Boursicaud et al., 2016; Le Coz et al., 2016; Dramais et al., 2011). This trend is expected to continue as technology such as smartphones and unmanned aerial vehicles become more common (Perks et al., 2016).

In this context, the floodX project was launched to create a comprehensive dataset documenting urban flood events in a controlled environment, in which conventional sensors are complemented with alternative sensors, namely video cameras and temperature probes. Three key points make the floodX project stand out. First, it is the largest scale controlled experiment examining the urban pluvial flood phenomenon (compare Fraga et al., 2015; Hakiel and Szydłowski, 2016; Testa et al., 2007). Second, it has a high density of sensors that cover the majority of the components of the hydraulic system, so a comprehensive picture of the flood events can be gained. Third, the sensors used are a combination of state of the art conventional sensors, such as a magnetic-inductive pipe profiler and radar-based flow measurement, and novel sources of data, such as temperature probes for detecting manhole overflow and surveillance cameras for monitoring water levels and overland flow velocities.

The data collected in the floodX project can enable or facilitate new research opportunities, such as:

- Development of new flood measurement concepts on the basis of temperature sensors, particle image velocimetry, and image interpretation.
- Development of new flood modelling and flood model calibration methods.
- Gaining insights into sensor performance during flood events.
- Benchmarking of urban flood models.

If successful, these developments will improve the accuracy of urban flash flood modelling, thereby increasing the effectiveness of urban flood management services, such as flood forecasting, which in turn reduce the vulnerability of society to this natural hazard.

## 1.2  Paper structure

The following sections contain essential information for understanding and using the data produced from the conducted flash flooding experiments. In section 2, the hydraulic network of the experimental setup is presented and its general hydraulic behaviour is described. In section 3, an overview of the sensor network is given. In section 4, the experimental procedure for executing the experiments is presented. In section 5, the preprocessing code used to transform, clean, and format the data for future analysis is introduced. In section 6, recognized weaknesses and limitations in the data are exposed, and some of the lessons learned in the course of the project are shared.

## 2  Hydraulic network

The experiments took place at a flood response training facility at a military village in the Canton of Bern, Switzerland (Figure 1). This facility was used for a period of five days, during which the whole experimental setup was installed, used, and dismantled. The facility is fed by a reservoir holding 450m$^3$ of water that is 10 m above facility and is continually filled from surrounding groundwater at low but relatively stable rate. The facility consists in a floodable area of around 500 m$^2$ with a





maximum elevation difference of 2.9 m. It contains a small construction with a basement and has a configurable drainage system with multiple drainage points and manholes. The floodX documentation package (doi: 10.5281/zenodo.248735) contains the detailed construction plans of the facility.

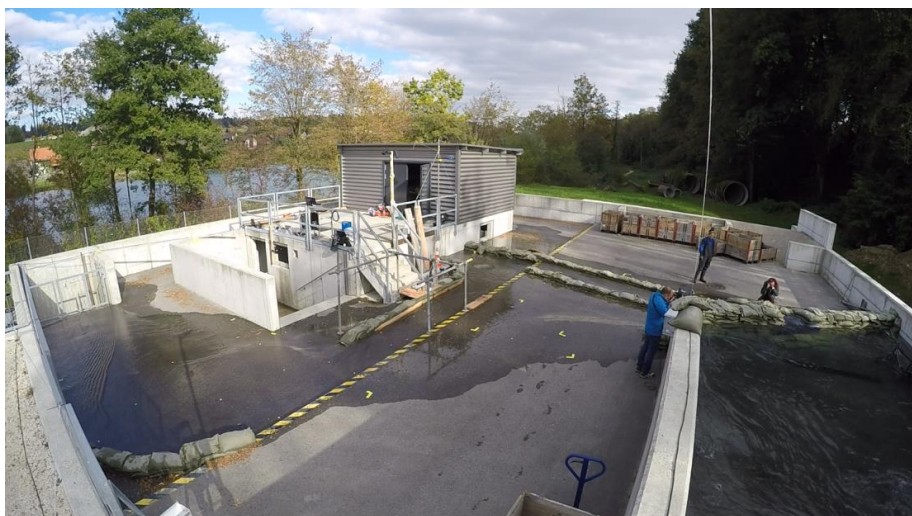

**Figure 1: Setup of floodable area during one of the experiments.**

The existing hydraulic network was adapted in a way that would instigate dynamic behaviour and increase the response time of the floodable area, despite the small size of the floodable area. To this end, sandbag walls were used to dam and channel water (Figure 1), and valves of the sewer network were configured so that the system would display the following phenomena relevant

10 to urban flooding: indoor and outdoor ponding of water, drainage into sewer inlets, overflowing of a surcharged manhole, and overland flow of one-dimensional and two-dimensional character. The resulting hydraulic network is depicted schematically in Figure 2. The locations of the hydraulic components can be found in the floodX documentation package (doi: 10.5281/zenodo.248735). Additional measures were undertaken, such as pipe extensions to encourage laminar flow, to improve the difficult measurement conditions caused by the relatively small scale of the flood facility.





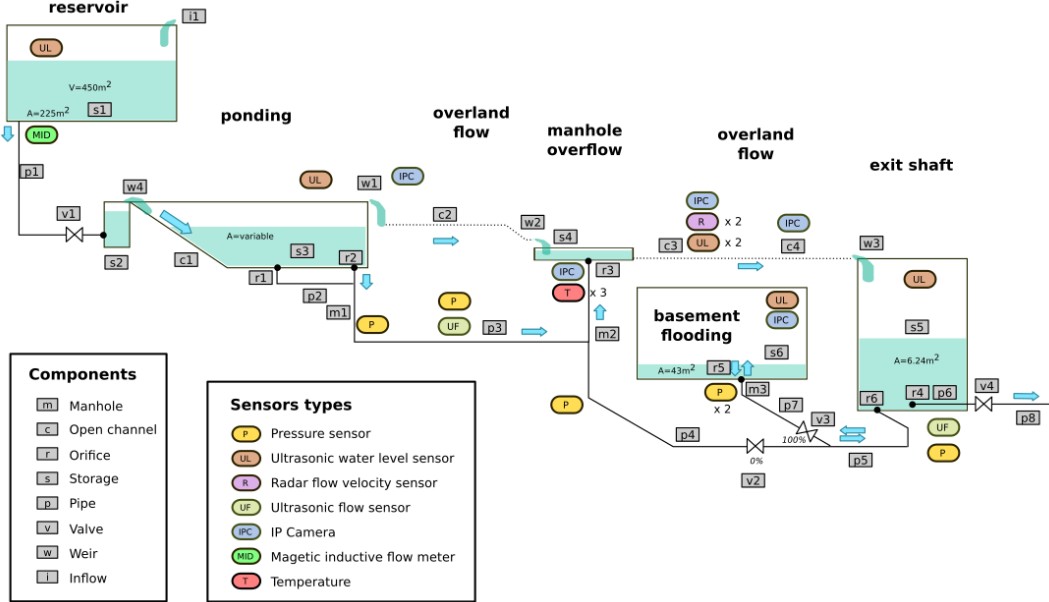

**Figure 2: Schematic representation of hydraulic network, including the labels of the main hydraulic components and the locations of vaious sensor types.**

The hydraulic system operates in the following way: water flows into the floodable area from reservoir s1 through pipe p1 into a small shaft s2. Rapidly, shaft s2 fills with water and overflows (weir w4) through a grating onto a ramp c1 that leads to a storage element s3. Storage s3 is caused by a dam constructed with sandbags on the opposite side of the ramp (see Figure 1), over which water can spill at weir w1 and into an open channel c2. Two orifices (r1 and r2) drain storage s3 into manhole m1, from which the water can flow to manhole m2 through pipe p3. The natural exit of manhole m2 is pipe p4, but valve v2 at the extremity of

pipe p4 was closed except for a slow leak during the experiments, causing manhole m2 to overflow through its opening r3 into a small storage area s4. Storage s4 is also the outlet of channel c2, which originates from the dam overflow w1. Storage s4 drains through open channel c3 (well defined channel walls) and channel c4 (no walls) into shaft s5, at the base of which there are two orifices. The first orifice (r4) leads out of the floodable area through pipe p6, valve v4 and pipe p8. The second orifice (r6) leads to a manhole (m3) in the basement of the small building through pipe p5, valve v3 and pipe p7. Because valve v4 was partially

closed, water could build up in the exit shaft s5 and cause manhole m3 to overflow through its opening r5 and into the basement s6. These hydraulic components are graphically represented in Figure 2 and their properties can be found in the floodX documentation package (doi: 10.5281/zenodo.248735).

## 3  Sensor network

A total of 18 sensor systems were installed to monitor flooding-relevant variables. For readability, the complete list of sensors,

including their mounting location and other relevant information can be found in the floodX documentation package (doi: 10.5281/zenodo.248735). The sensors used were not only state of the art equipment used in urban drainage monitoring, but also security cameras and temperature probes for monitoring surface flooding. The various types of measurement systems and their



respective measurement frequencies are provided in Table 1. Naturally, some sensor systems were composed of multiple sensors, for example a Radar flow measurement system not only measures surface velocity but also water level. Additionally, the data from certain sensors were logged with multiple data loggers in order to increase the sampling frequency (see floodX documentation). For this reason, a clear nomenclature was developed for designating data (square brackets indicate

interchangeable text):

<b>[p1]_[q]_[mid]_[endress]_[logi]</b>

- **p1**: location of measurement (see Figure 2 for complete list of hydraulic components, each of which is a possible location)
- **q**: physical variable measured by sensor. Can be one of: **q** (flow), **h** (water level), **v** (surface velocity), **t** (temperature)
- **mid**: measuring principle of the sensor, one of: **mid** (magnetic-inductive), **p** (barometric), **us** (ultrasound) **radar** (Doppler radar)
- **endress**: make of the sensor, one of: **endress (for Endress+Hauser)**, **maxbotix**, **nivus**, **hach**, **onset**, **instar**
- **logi**: optional appendix for logging method or number to distinguish multiple sensors at same location; see floodX documentation for details

This formula applies to all sensors, with exception of the video cameras and temperature sensors for which the measurement principle is lacking.

## 4   Experimental procedure

For the experiments, artificial flood events were created by partially opening valve v1, thereby letting water out of the storage reservoir s1 through pipe p1 and into the shaft s2, from where it spills out into the floodable area. It is advised to refer to Figure 2

or to the plans provided in the floodX documentation package (doi: 10.5281/zenodo.248735) in order to gain familiarity with the numerous hydraulic components of the experimental setup.

### 4.1   Initial conditions of the hydraulic network

To ensure that manhole m2 would overflow, the valve v2 was closed for each experiment. However, this also meant that water could not properly drain from pipes p3 and p4 at the end of a flood event. Most often, before the start of an experiment, valve v2

was opened and pipes p3 and p4 were allowed to drain. However, in a few cases, the water was left in pipes p3 and p4. The consequence of pipes p3 and p4 being full is that the system would respond much faster to inflow, thus resulting in manhole m2 overflowing very rapidly. The experiment metadata file contained in the floodX documentation package (doi: 10.5281/zenodo.248735) documents the status of pipes p3 and p4 for each experiment.

Valve v4, which allows the whole facility to be drained, was fully opened between each experiment to ensure that any blockage

in the valve would be flushed away. These two valve openings can be identified in the full dataset (preprocess the data with the option export_selection='all_data') as peaks in flow and abrupt changes in water level in between experiments.

### 4.2   Experiment execution

Artificial flash floods were created by manually opening valve v1 at the entry of the flood facility. The hydrographs used for the experiments were based on simple step-like functions rather than realistic-looking hydrographs. There are several justifications

for this:




- The valve controlling the inflow to the system was controlled manually and therefore the flow commands had to be simply defined.

- Well-defined shapes make the experiments more reproducible.

- The presence of a high storage node (s3) at the very start of the hydraulic network dampens high-frequency variability
of flow into the system. The extra effort to produce such variability was therefore not justified.

The individual experiments were planned to be expedient as possible, so as to optimize the experimentation time. For this reason, experiments were stopped after the main dynamics of the system stabilized, even though water often remained in some pipes as well as in the exit shaft s5. The experiments range from a scale of 2.5 to 77 m$^3$ discharge volume, and last between 5 minutes and one hour. The environmental conditions in which the experiments were carried out include overcast skies, direct sunlight, windy
conditions, and night. A complete list of the experiments conducted is available in the metadata/experiment_list.csv file provided in the floodX Datasets package (doi: 10.5281/zenodo.236878). Because there is value in observing a range of system responses (e.g., occurrence of dam overflow), the experiments were planned with references to such responses, instead of a purely quantitative description of the hydrograph. As an example, a snapshot of the data for one experiment can be seen in Figure 3 below.

**4.3    Variable quality of flood experiment data**

The first flood experiments executed were inevitably fraught with problems in the experimental setup and data logging, and so they cannot all be compared with experiments carried out after changes were made to the experiment to fix the problems. Each experiment has been tagged according to its level of quality in the experiment metadata file, which can be found in the floodX Datasets package (doi: 10.5281/zenodo.236878).

**5     Preprocessing of data**

To facilitate the use and reuse of the flood datasets, not only the raw data that was collected but also a preprocessed version of the data and the Python code used to perform the preprocessing is provided. In the following, the transformations that can be applied to the data with the code are briefly introduced. It is suggested that the interested reader investigate the processing code directly for insight on how exactly the transformations are implemented. Importantly, flood image and video data is not modified
but provided as collected (doi: 10.5281/zenodo.249053 and 10.5281/zenodo.232460, respectively).

**5.1    Reading sensor display images with optical character recognition**

Because of the low logging frequency of the minilog data loggers, which was only discovered in the middle of the experiment campaign, high frequency measurements for sensors p1_q_mid_endress, m1_h_p_endress, m2_h_p_endress, and m3_h_p_endress were only possible by taking images of the sensor or logger displays. These images are provided in the floodX
Data Logger Images package (doi: 10.5281/zenodo.231187).

To process the images into text data, the preprocessing script makes use of a well-known implementation of optical character recognition (OCR) called tesseract-ocr (Smith et al., 2007), which is called from Python with a wrapper (pytesseract[2]). The Python script first clips out the area(s) of interest from the image before processing it with OCR. Since the position(s) of the

---

[2] https://pypi.python.org/pypi/pytesseract



display(s) sometimes changed between recording sessions, the images are grouped by session and are associated with a settings file that indicates the bounding box and rotation of the sensors.

Because the image exposure was sometimes too large, some of the images are an overlay of two values on the display. In some cases, this causes the OCR algorithm to falsely interpret the value and an error is produced. In some cases, this error is very great

and the outlier value can be removed with a simple threshold during data preprocessing. In the future, more advanced filters could be implemented for the data.

While the tesseract-ocr solution is effective for the images of p1_q_mid_endress to log water flow in pipe p1, it does not work with images with the m[1/2/3]_h_p_endress for water levels at manholes, possibly because of the image quality and the small size of the displays that disturb the image segmentation process. No solution for reading pressure measurement images was

invested into, since the variation of water level at the manholes is not as rapid as it is for flow in pipe p1. Just in case, additional image material for the pressure sensors has been provided in the form of HD video material (floodX Data Logger Videos package, doi: 10.5281/zenodo.235899).

### 5.2 Preprocessing time series

For time series data stored in text format, the following transformations are applied:

- **Consolidation**: Data for a single sensor but stored in multiple source files are consolidated into a single file. The paths to the source files are contained in the metadata.csv file. The specific formatting of each source file is specified in metadata.csv file.

 - **Sorting**: Since the data is consolidated from multiple files, each data entry must be sorted in chronological order.

 - **Formatting**: Each data logger uses a particular formatting system for the csv formatting, the date and time, and for null
values. These formats are homogenised according to the user preferences defined in settings.py. However, all null values are simply removed from the data.

 - **Offsetting**: Ultrasonic water level sensors measure the gap between the water and the sensor. To obtain the water level, this value has to be offset by the distance between the ground and the sensor (ground_level value in metadata.csv).

 - **Time shifting**: Delays and time zone differences of logger clocks are corrected (time_shift in metadata.csv).

- **Removal of extreme values**: Impossible values such as negative water levels and extremely large values due to OCR misreading of the sensor displays are adjusted or removed, respectively (max_valid_value and floor_value in metadata.csv). Some data loggers, like those from Nivus and Hach, remove impossible values automatically.

 - **Segmentation of data by experiment**: The processed data can be exported either grouped by experiment, for modelling, or ungrouped, for viewing and exploring. The data grouped by experiment are cropped to exclude time
between experiments, e.g., when the system is flushed. Export grouping preferences are set by the export_selection option in settings.py.

 - **Formatting for time series database**: For viewing and exploring the data, we recommend using CrateDB (https://crate.io/) for a time series database and Grafana (http://grafana.org/) as a graphical dashboard. To import the data into CrateDB, the data must be formatted in a JSON-like fashion. If the write_crateDB option is set to True, then
the data will be written in the required JSON format as well. Figure 3 is a screenshot from the Grafana dashboard that is linked to a CrateDB database containing preprocessed flood data.



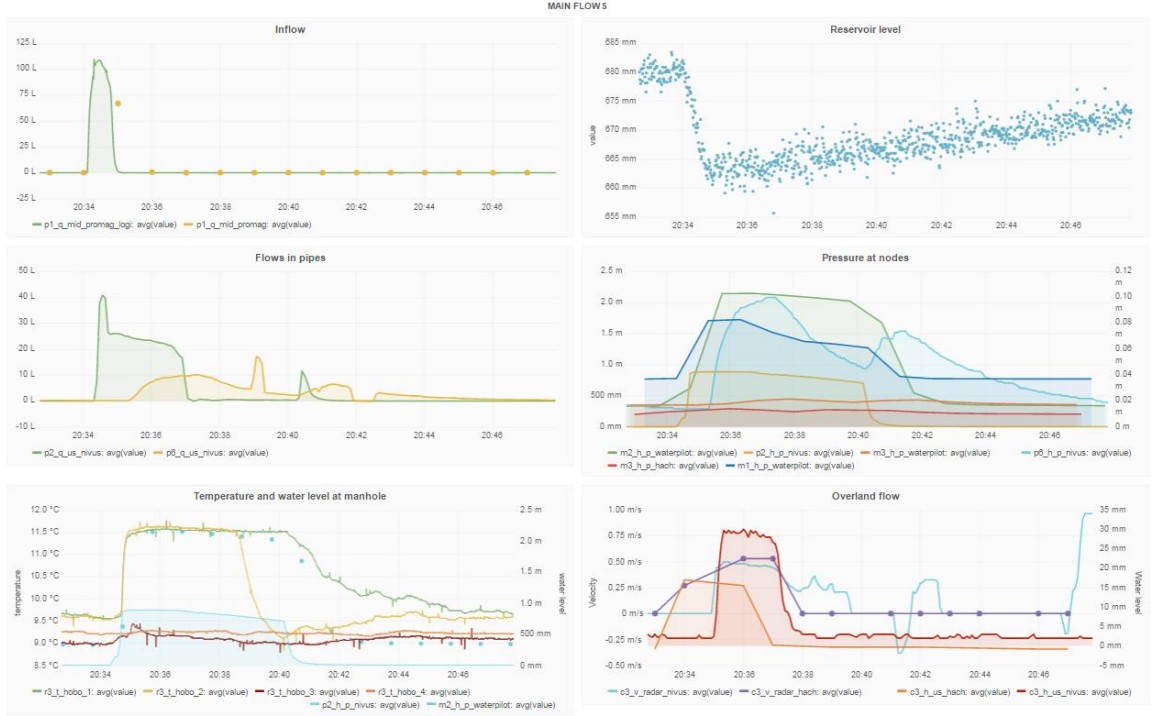

**Figure 3: Extract of data for one experiment, for a selection of sensors. The hydrograph can be seen in the upper left chart in green.**

### 5.3 Development of the preprocessing code

To further enhance the usability of the data, the code will be continually improved. The live code is available at
5   https://github.com/eaw-sww/floodx_data_preprocessing.git. The following additions are anticipated but not scheduled:

- Generate output files at a certain temporal resolution (aggregate or interpolate the data – holes will also be filled in the process.
- Implement a smarter OCR algorithm that can interpret the pressure data readings.
- Implement filtering methods to remove outliers from data and reduce noise.
10  - Implement an algorithm to correct drifts in sensor measurements.

### 6   Weaknesses of the data and lessons learned

### 6.1   Issues with data logger synchronization

For a majority of the sensors, an offset in the time records was observed. While this offset was usually constant and therefore amendable, notable synchronization issues were identified for the following sensor systems.



### 6.1.1 IP cameras

The internal clocks of the cameras were not synchronized, apart from CAM2, which was pointed on manhole m2. Temporal alignment of CAM2 was confirmed thanks to a comparison of the camera images with the pressure sensor p3_h_p_nivus and the three temperature sensors r3_t_onset_[1/2/3], using overflows of manhole m2 as markers.

Because the cameras were started and stopped with a Windows Batch script, all cameras started recording at the same moment in time (within one second at most). By comparing the video file times of the recordings, the relative misalignment of the camera's internal clocks could be quantified (Table 2). Visual investigation of moments in the video material confirms the related misalignments. The offsets were only determined for the last two days (, since the experiments conducted previously were not judged to be of sufficient quality to make the effort worthwhile. The interested researcher can investigate other recording
sessions to determine missing temporal offsets as needed.

### 6.1.2 Overland flow sensors at channel c3 for water depth and velocity (c3_h_us_nivus and c3_v_radar_nivus)

A comparison of the video material from camera c3_cam3_instar with measurements from overland flow measurements from sensors c3_h_us_nivus and c3_v_radar_nivus revealed a discrepancy in the time of the logger of these two devices. The logger time is consistently found to be between 5 and 16 seconds ahead of reference time. Given the logging frequency of 5 seconds of
the Nivus PCM Pro logger used for c3_h_us_nivus and c3_v_radar_nivus, it is assumed that the variability of the offset is largely due to sampling errors. A fixed offset of 12 seconds was assumed for preprocessing the data.

### 6.2 Issues due to drift in sensor measurements

Certain sensors displayed a certain amount of instability or drift in their measurements.

- **s1_h_us_maxbotix_2**: This ultrasonic water level sensor was unknowingly moved during the experiments. It was
moved on 5.10.2016 to location s3 where it is named s3_h_us_maxbotix, to measure the water level behind the dam.

- **s1_h_us_maxbotix_1**: This ultrasonic water level sensor was left in the reservoir for the duration of the experiments. Waves on the surface of the water can be seen in the signal. The amplitude of these waves increases as the water level in the reservoir decreases.

- **s3_h_us_maxbotix**: During periods in which the dam is empty, a drift in the sensor measurement can be noticed
(Figure 4). However, since the drift was found to not follow a linear trend, no simple corrective action was possible. It is hypothesized that the drift is linked to changes in ambient temperature and possibly direct solar radiation on the sensor body.



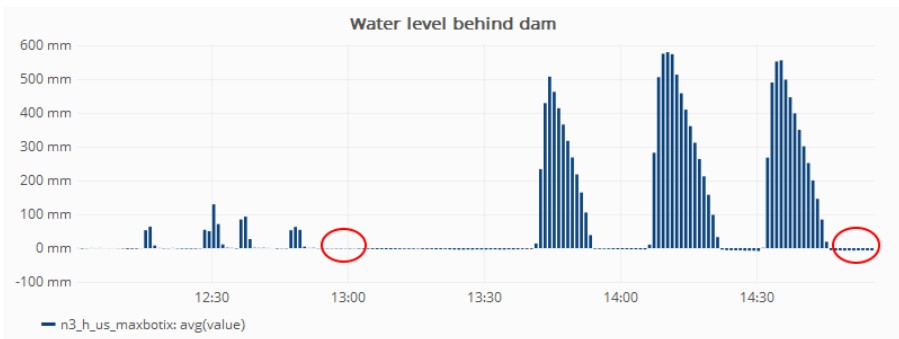

**Figure 4: Drift in water level measurement thought to be associated with temperature of sensor. The "empty" water level can be seen decreasing over time.**

- **c3_h_us_nivus**: This ultrasonic water level sensor displays a drift that correlates with ambient temperature. The offset appears to become visible as ambient temperature (measured by r3_t_onset_4) falls below 11°C. It is suspected that above this temperature the offset is not visible anymore because the device data logger rounded negative water level values to zero.

- **c3_v_radar_nivus**: During absence of flow, the radar sensor produced recurring outlier values over periods of up to around a minute. Inspection of the video of camera r3_cam2_instar showed that this problem occurred when a person walked on the crossing in front of the radar. When flow is present, however, the passage of people on the crossing does not appear to effect the measurement.

- **p1_q_mid_endress_logi**: Significant outliers can be found in the flow data. These data values were read using optical character recognition from images of the flow meter display. Invariantly, there are errors in the interpretation, especially when the image was taken at the same moment at which the value was being updated on the display.

- **s5_h_us_maxbotix_2**: This ultrasonic water level sensor was moved from another location in exit shaft s5, where it was called s5_h_us_maxbotix_1, because at the first location the signal was disturbed by the water falling into the shaft over weir w3. As in the case of other ultrasound-based devices, s5_h_us_maxbotix_2 seems affected by temperature, since a drift appears at around the same time as for s3_h_us_maxbotix. However, it is also possible that the apparent drift is simply a change in the amount of water remaining at the bottom of shaft s5 in between experiments.

- **s6_h_us_maxbotix**: Also for this ultrasonic sensor, the measurement seems affected by temperature. Additionally, a strange inertial effect is visible after flooding occurs (Figure 5).

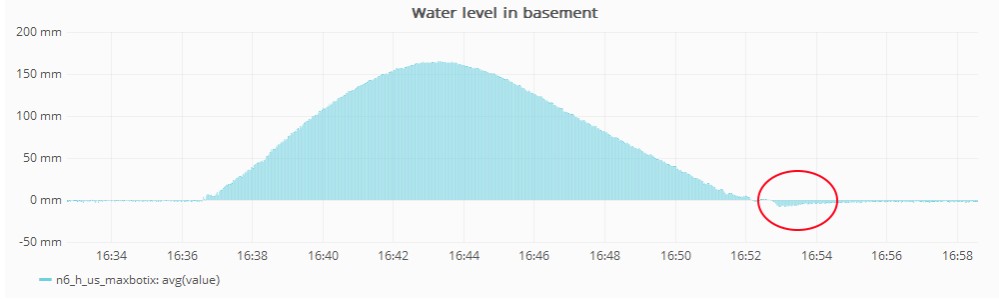

**Figure 5: Measured water level in basement s6 presents abnormally low values after flooding.**




### 6.3 External disturbances to the measurement

Contactless sensors like the radar sensors and ultrasonic water level sensors are sensitive if foreign objects come too close. For example, the radar systems were placed too close to the stairs of the building in the centre of the floodable area, so people walking by were in range of detection. While water is flowing on the ground this extra signal is filtered out, but if there is no water flowing then these foreign signals are not filtered out but rather accentuated. Such situations are easy to identify because the water level measurement remains unchanged while the velocity spikes.

### 6.4 Environmental forces

At times, there was moderate wind which caused two major disturbances. First, a large quantity of leaves was released from the surrounding trees, leading to clogging of valve v4. For this reason, the valve was opened to a larger setting to encourage flow and discourage blockage. Additionally, the valve was opened between experiments to increase the flow rate and flush the system. Second, the wind caused any free-standing sensors to shake, making it necessary to anchor the security cameras and ultrasonic sensors with ropes.

### 6.5 Incomplete control of facility infrastructure

In the data, the water level in manhole m2 can be seen sinking (from 17:42 in Figure 6), despite there being no expected outlet, since all valves are supposed to be closed.

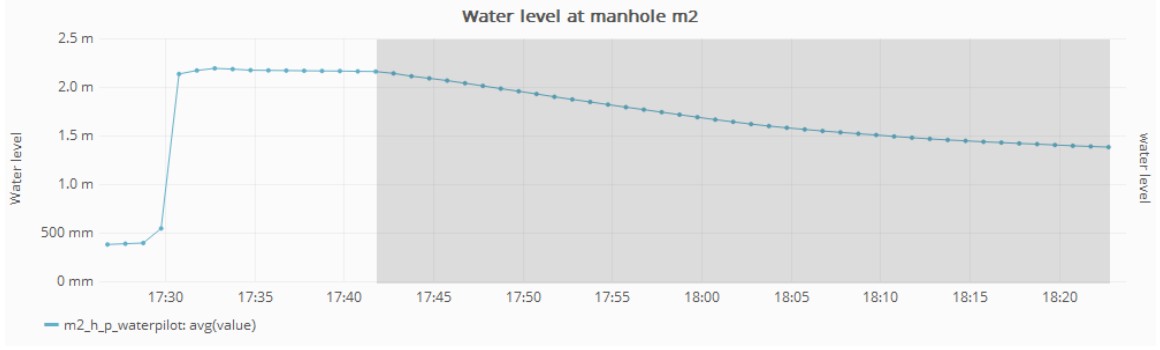

**Figure 6: Signs of leakage in manhole m2. During the period marked in grey, the water level in manhole m2 falls despite their not being any expected outlet for the water.**

### 6.6 Water volume balances

It was observed during data analysis that the flow measured entering the system (through pipe p1, measured by p1_q_mid_endress_minilog) did not match that exiting the system (through pipe p6, measured by p6_q_us_nivus). Over the whole period of experimentation, a loss of around 20% is registered. Considering how the losses correlate with event size (red circles in Figure 7), the relative losses are large and very variable for small events, but there is a clear trend in which the relative error decreases to below 10% for large events. There are multiple reasons that could account for the discrepancies between the flows in and out of the system. First, residual water was present in the pipes of the system at the beginning and end of the experiments. Since the volume differences due to residual water should be independent from the event size and can be both negative and positive, this factor could be responsible for the high variability of smaller events (below 15 m$^3$). This factor cannot explain all variability because on average it would not affect the balance very much, and the relative error should become




negligible for large events. Second, it is possible that the measurement in pipe p6 was erroneous, possibly because of difficult flow conditions, incorrect calibration or incorrect installation. It is important to note that the flow conditions were rather sub-optimal in pipe p6: There were frequent transitions between partially filled and filled pipe conditions, a risk of air getting stuck in the pipe, a partially open valve at the end of the pipe, and a short stabilizing zone. The resulting measurement error could explain

volume differences in a range of 0-10% such as those that follow the trend line for medium and large events. However, there are two large events in Figure 7 (at approx. 45 and 65 $m^3$) that are outliers to the trend. Third, it is possible that the water is escaping from the network, either through valves that are not fully closed or actual leaks in the network. Fourth, the sandbag walls were not completely impermeable, so water was able to escape from the closed hydraulic network depicted in Figure 2. However, none of these reasons can fully and satisfactorily explain the apparent water loss for the two outliers mentioned above.

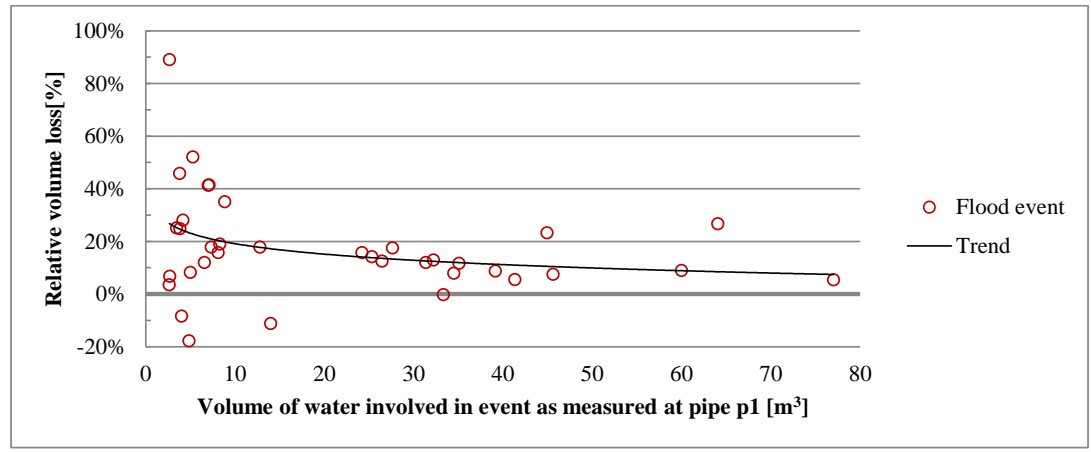

**Figure 7: Relative difference between flood volumes measured at entry and exit of floodable area, aggregated by flood event and plotted according to event size.**

### 6.7 Gaps in the data

Equipment manipulation errors and data management issues led to a few gaps in the data. This has been observed for the Nivus
radar system mounted in channel c3 in experiment 7, sensors m1_h_p_endress and m2_h_p_endress for experiments 14 to 23, and the temperature sensors in experiment 24. This information is also available in the experiment metadata file in the floodX Datasets package (doi: 10.5281/zenodo.236878).

### 6.8 Lessons learned from data collection

The novelty and complexity of the floodX project provided the occasion to learn several practical lessons first-hand. In hindsight,
many of the missteps could have been avoided and the authors would be happy to share their experiences with the interested reader. The following lessons seem especially important:

- Synchronization of data loggers must be performed manually and at an appropriate temporal resolution.
- Unpredictable environmental conditions like wind can influence experiment execution and should not be underestimated.
- Similarly, wind and direct sunshine can affect contactless sensors like ultrasound-based water level measurement. Direct sunshine on a device can increase the local temperature and lead to over- or under-compensation of the measurement.



- New types of tasks should be planned with threefold the time that would be needed under normal circumstances.
- Checklists should be prepared and used to make sure equipment is functioning before each experiment.
- Performance of data loggers can be as important as that of the sensor, depending on the time scale of an experiment's dynamics.

## 7 Data availability and license

Six data packages are provided through the Zenodo data repository as described below, in open-access and with liberal licenses (see Table 3 for a summary). It is important to note that while the time series data (floodX Datasets) is supplemented with code to preprocess as described above, the videos and images of the flooding (floodX Flooding Videos and floodX Flooding Images) are provided as-is. Data can be unpacked with 7zip[3]. A brief description of the data packages follows.

### 7.1 floodX Datasets

This package contains all collected data that is available in a text format, as well as useful code for preprocessing data. The following subfolders are contained within "floodX Datasets":

- **metadata**: Contains metadata for the sensors used, the datasources (sensor + location + data logger), flooding experiments and image files for optical character recognition.
- **data_raw**: Contains the raw text format data, organized by sensor.
- **code**: Contains processing code for making the raw data more usable for visualization and modelling. The code is maintained in the following github repository: https://github.com/eaw-sww/floodx_data_preprocessing.git
- **data_ocr_result**: Contains data read from sensor display images using the processing code.
- **preprocessed**: Contains the preprocessed data.

### 7.2 floodX Flooding Videos

This package contains archives of videos of the flooding taken with surveillance cameras. The videos are grouped by camera and by recording sessions.

### 7.3 floodX Flooding Images

This package contains pictures taken during the flooding experiments. The cameras used to take the pictures had shifted timestamps, so the README provides the time shifts of the cameras to the reference times.

### 7.4 floodX Data Logger Images

This package contains archives of pictures taken of sensor displays in order to record measurements at a higher frequency than certain data loggers allowed. The data for the pressure sensors is stored in multiple archive files that must be assembled during unpacking. The images are grouped together because sometimes the camera had to be moved between experiments, thereby changing the position of the display in the image. Each image group contains a settings file that indicates the location of the display(s) in the images of the group. With this information, the sensor reading can be automatically extracted from the images.

---

[3] http://www.7-zip.org/



### 7.5 floodX Data Logger Videos

This package contains videos that constitute an alternative to the three minilog logging devices, which were designed to save only one water level value a minute. The videos provide high-quality readings of the logger displays, and values can be read more frequently. The image quality may be superior to that of the Data Logger Images

### 7.6 floodX Documentation

This package contains an archive with material documenting the flood facility and the sensors that were used in the experiments.

- Facility construction plans (provided as-is, without any guarantee for geometric accuracy).
- Plan of floodable area, including labels for hydraulic components and sensors as defined in this paper.
- Information regarding individual sensors and dimensions of hydraulic elements such as pipes and storage elements.
- Photos of the experimental layout.

## 8    Conclusions

The flood experiments described in this paper stand out from similar urban flood experiments thanks to the relatively controlled conditions and the diversity of sensor systems involved. While not void of blemishes, the data generated in the floodX project so far holds significant potential to support urban flooding research, especially for the exploration of alternative measurement strategies for the quantification of urban flood phenomena. In particular, the published datasets constitute a valuable starting point for investigating large-scale particle image velocimetry in urban environments, computer-vision based water level estimation, and simple temperature-based flooding detection. Indeed, the need for flood monitoring data has often been repeated, and alternative data sources could be the key to providing such data. Additionally, the datasets provide a unique opportunity for benchmarking and developing faster and more reliable urban flood models. In the face of climate change, the potential improvements to urban flood monitoring and modelling expedited by these data will contribute to the integrity of urban infrastructure and the populations that rely on them.

## 9    Author contributions

Matthew Moy de Vitry was lead for the project design and execution, as well as for drafting the paper. Simon Dicht was the lead technician, responsible for the acquisition and installation of instrumentation. João P. Leitão was principle investigator of the project, providing valuable support in the orientation and coordination of project execution and paper drafting. All authors were involved in reviewing the manuscript.

## 10    Acknowledgments

The authors would like to thank the following people for their valuable counsel for conceptual design of the experiments: Tobias Doppler, Frank Blumensaat, Andreas Scheidegger and Kris Villez. Additionally, great assistance was provided in execution of the experiments by Christian Ebi, Alex Hunziker, Lena Mutzner, Joerg Rieckermann, Andreas Scheidegger, Luis M. de Sousa and Omar Wani. The project was made possible thanks to the cooperation of armasuisse and the training village facility



managers Roland Rickli and Michael Gehriger. Furthermore, the authors thank Stebatec AG and Nivus AG for their collaboration in providing sensor equipment.

## 11   Disclaimer

This project was financed by the Swiss National Science Foundation under grant #169630. The authors declare no conflict of
interest.

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



**Tables**

**Table 1: Sensor types installed for flood monitoring.**

| Sensor Type | Variables measured | Maximum temporal resolution | Number of sensors |
|---|---|---|---|
| Pressure sensor | Water level | 10 seconds* | 4 |
| Temperature sensor | Manhole overflow | 1 second | 4 |
| Ultrasonic distance sensor | Water level | 200 milliseconds | 4 |
| Radar system | Surface flow velocity and water level | 5 seconds ** | 2 |
| Ultrasonic flow sensor | Flow and pressure in partially filled and filled pipes | 5 seconds | 2 |
| Security Camera | Surface flow velocity and water level | 25 frames/second | 5 |
| Magnetic-Inductive pipe profiler | flow in filled pipe | 1 second* | 1 |

**\* The data recorded directly by the sensor data logger has a resolution of one minute, but a greater temporal resolution can be acquired by interpreting images of the data logger screen (see Section 5.1).**

5  **\*\* Two systems were installed at the same location. One has a temporal resolution of one minute and the other of five seconds. See floodX Documentation package for more information.**

**Table 2: Temporal offset of cameras during the last two days of the floodX experiments.**

| | Recording start times | | | Relative offset to reference time | | |
|---|---|---|---|---|---|---|
| Recording session name | 161006A | 161006B | 161007A | 161006A | 161006B | 161007A |
| **cam1** | 12:02:38 | 19:58:22 | 11:08:33 | 00:00:03 | 00:00:02 | -00:00:01 |
| **cam2** | 12:02:35 | 19:58:20 | 11:08:34 | 00:00:00 | 00:00:00 | 00:00:00 |
| **cam3** | 12:02:45 | 19:58:29 | 11:08:41 | 00:00:10 | 00:00:09 | 00:00:07 |
| **cam4** | 12:02:32 | 19:58:16 | 11:08:23 | -00:00:03 | -00:00:04 | -00:00:11 |
| **cam5** | 12:02:25 | 19:58:09 | 11:08:20 | -00:00:10 | -00:00:11 | -00:00:14 |

10  **Table 3: Data packages associated with the floodX project.**

| Package Name | DOI | License |
|---|---|---|
| floodX Datasets | 10.5281/zenodo.236878 | Creative Commons Zero 1.0* |
| floodX Documentation | 10.5281/zenodo.248735 | Creative Commons Attribution |
| floodX Flooding Videos | 10.5281/zenodo.232460 | Creative Commons Zero 1.0 |
| floodX Flooding Images | 10.5281/zenodo.249053 | Creative Commons Attribution |
| floodX Data Logger Videos | 10.5281/zenodo.235899 | Creative Commons Zero 1.0 |
| floodX Data Logger Images | 10.5281/zenodo.231187 | Creative Commons Zero 1.0 |

**\* The code included in the floodX Datasets package is under the MIT license.**