# Peer review of "floodX: Urban flash flood experiments monitored with conventional and alternative sensors"

_Earth System Science Data, 2017_

## Short Comment (SC1) · 12 Jun 2017

The submitted manuscript presents a data set of a five days experiment conducted on an experimental 50 m2 platform aimed at reproducing urban floods. The concrete platform is connected with an underlying pipe and a reservoir representing a basement through a manhole. Various input discharge scenarios are tested and the induced surface and pipe flows are measured through a series of sensors : pressure sensor, ultrasonic water level, ultrasonic flow, radar, cameras, magnetic flow meter and temperature. The motivation of this experimental setting is to provide data that could be used to calibrate and test urban flooding numerical models and to demonstrate the usefulness of some additional sensors in cities such as temperature probes to detect manhole overflows or cameras providing films that can be interpreted to estimate

water levels or velocities (PIV). The presented work and database does not appear sufficiently advanced to be published in ESS.

First, the authors do not demonstrate the real usefulness of the produced data. They do neither show comparisons between measured and simulated data, nor try to use the additional proposed data (temperature or films). No attempt of surface velocity retrieval through PIV methods based on films is for instance proposed.

Second, the authors are honest, revealing the limits and uncertainties affecting their database. They often go too much into the details in describing these uncertainties or errors. Some described errors are basic - time shifts between sensors, sensitivity of the measurements to temperature and even perturbations due to external elements (such as pedestrians perturbing radar measurements). Some of these errors should be avoided if possible during the experiments, other should be filtered out during data processing. The critical analysis of data sets left to the users should be limited. The users should be able to download what can be considered as high quality data. Some sentences are worrying in the text : "The offsets were only determined for the last two days (, since the experiments conducted previously were not judged to be of sufficient quality...", suggesting that the data set contains data of highly variable quality and that the authors let the users make their own selection. This is not really acceptable and compatible with the quality requirements of the data sets presented in ESS.

Third, the experimental setting is relatively impressive (volume, surface) but represents an extremely simple system : one single manhole. I have some doubts that such a simple experiment provides original and new data sets to test and calibrate urban flood simulation models. At least this should be illustrated in the manuscript.

Finally, the figures in the manuscript are hardly readable and should be significantly improved. An annotated manuscript is attached to this review.

Please also note the supplement to this comment:
http://www.earth-syst-sci-data-discuss.net/essd-2017-7/essd-2017-7-SC1-supplement.pdf

**Supplement:**

[revised manuscript text omitted]

---

## Referee Comment (RC1) · Anonymous Referee #1 · 13 Jun 2017

An interesting work. However, I have following queries:

1) The setup created by authors seems to be very idealized?.

2) How it can be compared in real flooding problems in a big city or urban areas?.

3) Generally flooding is by severe rainfall which is some what distributed. In the experiments, it is coming from reservoir.

4)how to identify important flood parameters such as 1) time of concentration; 2)time to peak?. 3) Max.flood depth; 4) area of inundation etc.

5) What is the "preprocessing code"?.

6) How effectively, this data can be used in flash flood studies?.

7) "A loss of 20% recorded" - how much it is for real cases?.

8) How effectively, the data generated can be used?.

9) Any separation between "channel flow and overland flow"?.

10)  What are the effects of area, intensity of rainfall, duration etc?.

11) In flooding, what is the use of particle image velocimetry?.

---

## Referee Comment (RC3) · Anonymous Referee #3 · 3 Jul 2017

**Review of paper ESSD-2017-7**
**floodX: Urban flash flood experiments monitored with conventional and alternatives sensors**
**by Moy de Vitry et al**

**General comments**

The paper describes experimental data sets of urban flooding experiments. The experimental set up covers a 500 m$^2$ area which provides data at almost a 1:1 scale. Such data sets are not very common and can be of high interest to test urban flood models in conditions close to field conditions. The authors should emphasize this point in their description of the data set. I agree with the authors that such a data set is very valuable to assess the validity of urban flood models.

If we refer to the scope of ESSD, it is stated that "Any interpretation of the data is outside the scope of a regular paper..... Any comparison to other methods is beyond the scope of regular data." Therefore, it is not expected to see comparisons of the data with numerical models in the paper, nor any complete description or interpretation of the data. However, ESSD expects good quality data sets and the description should be clear and complete enough to provide the relevant information useful for potential users. I believe the authors did not really succeed to reach this last goal for the following reasons:

1/ the experimental set up description provided in the paper is not clear and complete enough to really understand the experimental set up and the various components of the experiment. The photo provided in Figure 1 does not provide the location of the sensors and does not allow understanding how the experiment was conducted and how water was flowing. Figure 2 provides a schematic transverse view of the experiment but it gives the impression that it is only a connection of pipes and valves, and does not inform on the lateral extension of the experiment. My point of view is that the authors should provide one or more plan view of their experiment showing the interconnections between the various elements, the 2D extension of their experiment and of water flow and also the locations of the various sensors (and possibly the view angles of the cameras). The authors make reference to the floodX documentation data set that provides plans of the experimental set up, but I believe this information should be provided in the main paper, in a schematic manner, but providing information on the main elements of the experiments (reservoirs, pipes, valves, manholes, etc..), of the location of the sensors and of the main directions of the flow.

2/ in the abstract the authors mention 37 experiments, but the paper does not provide an overview of the content of those experiments. The reader should refer to the data set documentation for that. I believe that a summary table, providing the main features of the experiments (duration, input discharge, specific configurations, etc.) should be provided in the paper, together with information on the data reliability of each data set (see next point). The potential user of the data should have guidelines to determine which experiment is reliable and which one is relevant for his/her specific needs.

3/ the authors also provide the codes used to pre-process the data. They provide the raw and pre-processed data which I find a good point, as the potential users have the possibility to take the raw data and make their own pre-processing if they do not agree with the authors' one. However, the pre-processing data presentation is somehow puzzling for the reader. As the authors are honest and do not hide the problems encountered with their data, the reader is left with the idea that none of the data set is of enough quality. I think the authors should provide clearer information on the data set quality, and if some experiments are not reliable enough, they should consider removing them from the data set.

4/ the authors explain with quite details, the problems they encountered with their experiments. This is a good point for the future users. However, the presentation is not balanced enough and the reader ends its reading with serious doubts about the interest of the data set. I believe the authors should also spend some times explaining what makes their data set valuable for other users, and what the strengths of their data sets are. Figure 3 is a nice summary of the collected data, but should be commented more in details to explain what can be seen in the figure. In particular the use of the temperature sensors is presented as new, but this would gain being illustrated with some examples. I also believe the OCR treatment of the data loggers is an original application that contributes to the originality of the data set.

5/ the authors should try to get a final version of the paper only focused on the data set description. So they may provide additional information (such as file naming, sensors drift, ..) in a supplementary material.

6/ at the very end of the paper, the authors mention water balance mass errors of about 20%. This is a big issue that may compromise the usefulness of the data set, and could require further analysis (in particular when putting doubts on the validity of the p6 discharge measurements). Shouldn't it be possible to make additional verifications to see if the proposed explanation may hold?

In conclusion, I believe that, provided the authors better document the quality of the various data sets and better describe the experiments, their experiments provide valuable data sets at a 1:1 scale that are of great interest for the science community, in particular for evaluating urban flood models. The data set has also some potential to develop LS-PIV technique. However, in its present state, the data set presentation suffers too many weaknesses for the paper to be published. I recommend major revision of the paper, following the suggestions provided above and below, before possible publication in ESSD.

**Specific comments**
1/ Section 1.1. The authors may also consider the following references (Bazin et al., 2014; Mignot et al., 2013)
2/ P. 2, line 12 "that covers the majority of the components of the hydraulic system": be more specific, which components?
3/ Figure 1, p.3 line 13: the figure is not clear enough and we do not see the elements mentioned by the authors. As mentioned before, one or several plan view of the experiment is necessary for the reader to understand how the experiment was working and the water flow paths. This is all the more necessary that Figure 2 gives the impression that the experiment is only a connection of reservoirs and pipes.
4/ P.5 put the section on files naming in a supplementary material, so that it is accessible to the reader without needing to download the whole data set.
5/ P. 6: section 4.3: I suggest that the authors provide a summary table with the various experiments that were conducted, their main characteristics and information about data quality.
6/ Section 5.1: you could illustrate what the images look like by providing some examples.
7/ Figure 3 is a nice example of the collected data but it is hardly described and analyzed in the paper. The figure should be enlarged so that the authors can highlight the part of the curves that are providing useful information. Such a description would strengthen the paper, by providing an insight on how the data can be interpreted and used to the readers. In addition, the authors mention the novelty of the temperature sensors data, but they do not demonstrate it. An explanation/discussion of the corresponding figure would provide elements to really appreciate the interest of such data.
8/ P.9 line 19 " … was unknowingly moved.." . What does unknowingly means?
9/ Figures 4 and 5: the points the authors want to highlight are not very clear
10/ Section 6.4: the authors should provide information on the experiments that were affected by this problem (in the summary table that would be added to the manuscript)
11/ Section 6.6: among the difficulties mentioned by the authors, the water balance closure problem is certainly the most critical one. The authors propose several explaining factors, but none of them is fully convincing. In particular, problems with discharge measurements at the outlet are one plausible explanation. But could the authors make additional analysis to see if this explanation really holds?
12/ Figure 7: I don't understand what the "trend" line in the figure is.
13/ P.13, line 1: I don't understand what the authors mean with this sentence.

References
Bazin, P.H., Nakagawa, H., Kawaike, K., Paquier, A., Mignot, E., 2014. Modeling Flow Exchanges between a Street and an Underground Drainage Pipe during Urban Floods. Journal of Hydraulic Engineering, 140(10). DOI:10.1061/(asce)hy.1943-7900.0000917
Mignot, E., Zeng, C., Dominguez, G., Li, C.W., Riviere, N., Bazin, P.H., 2013. Impact of topographic obstacles on the discharge distribution in open-channel bifurcations. Journal of Hydrology, 494: 10-19. DOI:10.1016/j.jhydrol.2013.04.023

---

## Author Comment (AC1) · 19 Jul 2017

**AUTHORS' REPLIES TO RC1 COMMENTS**

**RC1.1**    *The setup created by authors seems to be very idealized?.*

**Authors:** While the experimental setup is very simple compared to a real urban catchment, it is still considerably larger and more diverse in urban flooding phenomena than comparable studies (Fraga et al., 2015; Hakiel and Szydłowski, 2016; Testa et al., 2007). Moreover, most of the phenomena that occur in an urban catchment are represented, e.g.: manhole overflow, basement flooding, ponding, overland flow, pipe flow, flow into sewer inlets.

**Changes:** none.

**RC1.2**    *How it can be compared in real flooding problems in a big city or urban areas?.*

**Authors:** The experiments were planned and conducted with two purposes in mind: First, the data is being used to develop and/or validate novel flood monitoring techniques that make use of tools such as Particle Image Velocimetry (PIV) or Deep Learning. Second, the data will be used to develop and test model calibration techniques. These two points are now stated explicitly in the manuscript.

The first application area is not affected by the size of the experiment. The second is affected but there are still a large number of model parameters that can be adjusted during the model calibration, e.g. surface and pipe roughness, weir coefficient and manhole discharge coefficient. The calibration insights gained with the experiment data are expected to provide urban drainage modelers with better understanding of the key points/ methods needed to accurately calibrate flood models of real urban catchments. Also, what the experiments lack in spatial extent they make up with the number of events and the density and diversity of sensors.

Based on the above points, the authors are therefore confident that the experiments and the collected data will also be useful for research about flood model calibration.

**Changes:** The following sentence has been added to the introduction : "In summary, the floodX data will be used for two distinct but related areas of urban flood research: (i) automatic interpretation of image data into useful information for flood monitoring and model validation, and (ii) development of flood model calibration methods with overland flow data to improve the predictive power of the models."

**RC1.3**    *Generally, flooding is by severe rainfall which is some what distributed. In the experiments, it is coming from reservoir.*

**Authors:** The authors acknowledge the reviewer comment. In the specific case of the conducted experiments the spatial distribution of rainfall of flood is not considered or represented. Nevertheless, the experiment remains representative of urban flood events for which flood water comes from a peri-urban catchment or even from an upstream urban catchment.

**Changes**: none.

*RC1.4* *how to identify important flood parameters such as 1) time of concentration; 2)time to peak?.3) Max.flood depth; 4) area of inundation etc.*

**Authors:** The flood parameters can be computed by analyzing the time series of the relevant sensors provided in the dataset. Two-dimensional information such as the area of inundation can be computed for pond s3 with the help of the water level and geometry.

**Changes**: none.

*RC1.5* *What is the "preprocessing code"?.*

**Authors:** The preprocessing code is the code used to transform raw data collected by the sensors into clean, good-quality and coherent datasets ready to be used by everyone in the scientific community interested in this topic. It is provided so that potential users can track all modifications undergone by the data and verify its quality. We provide a short description on page 9, lines 31-33.

**Changes**: none

*RC1.6* *How effectively, this data can be used in flash flood studies?.*

**Authors:** Please refer to the reply to comment 2.

**Changes**: none.

*RC1.7* *"A loss of 20% recorded" - how much it is for real cases?.*

**Authors:** We assume the referee is asking about the water loss for individual flood experiments. This can be obtained with help of the figure 7 page 12 of the original manuscript, in which each point represents one experiment.

**Changes**: Based on feedback from other reviewers the discharge data from pipe p6 has been removed from the datasets and this discussion has been removed from the manuscript.

*RC1.8* *How effectively, the data generated can be used?.*

**Authors:** Please refer to the reply to comment 2.

**Changes**: none.

*RC1.9* *Any separation between "channel flow and overland flow"?.*

**Authors:** We assume the referee is asking whether the terms are differentiated in the manuscript. The difference is that channel flow is a form of overland flow that can be assumed to be one-dimensional without introducing significant errors. Channel flow can be assumed in all pipes and channels except c4, where the flow is 2-dimensional. This is now specified in the floodX documentation, Table 5.

**Changes:** We added column to Table 5 in supplementary material.

**Table 5: Characteristics of open channels**

| Name | Max depth [m] | Length [m] | Surface | Flow | Shape | Width [m] | Start elevation | End elevation |
|------|---------------|------------|---------|------|-------|-----------|-----------------|---------------|
| C1 | 0.7 | 7.2 | Pavement | 1D | rectangular | 5.5 | 420.48 | 419.94 |
| C2 | 0.2 | 7.3 | Pavement | 1D | rectangular | 1 | 419.94 | 419.91 |
| C3 | No | 4.63 | Pavement | 1D | rectangular | 1.06 | 419.76 | 419.69 |
| C4 | No | 11 | Pavement | 2D | Ill-defined | variable | 419.69 | 419.47 |
* * *
**RC1.10**  *What are the effects of area, intensity of rainfall, duration etc?.*

**Authors:** The effects of area and intensity and duration of rainfall are represented by the intensity and duration of the inflow events.

**Changes:** none.
* * *
**RC1.11**  *In flooding, what is the use of particle image velocimetry?.*

**Authors:** The anticipated application of PIV is to quantify overland flow discharge during flood events in a cheap and robust manner, using existing surveillance infrastructure. This is now specified in the manuscript, page 2 lines 5 and 6. These data will allow urban drainage modelers to validate and calibrate urban flood models to improve model reliability.

**Changes:** The following sentence was revised: "In particular, optical methods such as large scale particle image velocimetry (LSPIV) can be used to estimate flood discharges and researchers have started leveraging social media and crowdsourcing to collect data for this purpose (Le Boursicaud et al., 2016; Le Coz et al., 2016; Dramais et al., 2011)"

**References**

Le Boursicaud, R., Pénard, L., Hauet, A., Thollet, F. and Le Coz, J.: Gauging extreme floods on YouTube: application of LSPIV to home movies for the post-event determination of stream discharges, Hydrol. Process., 30(1), 90–105, doi:10.1002/hyp.10532, 2016.

Le Coz, J., Patalano, A., Collins, D., Guillén, N. F., García, C. M., Smart, G. M., Bind, J., Chiaverini, A., Le Boursicaud, R., Dramais, G. and Braud, I.: Crowdsourced data for flood hydrology: Feedback from recent citizen science projects in Argentina, France and New Zealand, J. Hydrol., 541, 766–777, doi:10.1016/j.jhydrol.2016.07.036, 2016.

Dramais, G., Le Coz, J., Camenen, B. and Hauet, A.: Advantages of a mobile LSPIV method for measuring flood discharges and improving stage–discharge curves, J. Hydro-environment Res., 5(4), 301–312, doi:10.1016/j.jher.2010.12.005, 2011.

Fraga, I., Cea, L. and Puertas, J.: Validation of a 1D-2D dual drainage model under unsteady part-full and surcharged sewer conditions, Urban Water J., 0(0), 1–11, doi:10.1080/1573062X.2015.1057180, 2015.

Hakiel, J. and Szydłowski, M.: Interaction between storm water conduit flow and overland flow for Numerical modelling of urban area inundation, in GeoPlanet: Earth and Planetary Sciences, vol. none, edited by P. M. Rowiński and A. Marion, pp. 23–34, Springer International Publishing., 2016.

Testa, G., Zuccalà, D., Alcrudo, F., Mulet, J. and Soares-Frazão, S.: Flash flood flow experiment in a simplified urban district, J. Hydraul. Res., 45(sup1), 37–44, doi:10.1080/00221686.2007.9521831, 2007.

---

## Author Comment (AC2) · 19 Jul 2017

**AUTHORS' REPLIES TO RC2 COMMENTS**

**Replies to general comments**

**Authors**: We thank the reviewer for their time and effort, and for the many valid points that were raised in the detailed manuscript comments. The manuscript and data has been changed accordingly and is thereby greatly improved for clarity and reuse.

| | |
|---|---|
| **RC2.a** | *First, the authors do not demonstrate the real usefulness of the produced data. They do neither show comparisons between measured and simulated data, nor try to use the additional proposed data (temperature or films). No attempt of surface velocity retrieval through PIV methods based on films is for instance proposed.* |

**Authors**: The main purpose of the study was to generate good-quality data sets that could support the development of flood model calibration tools in a broader sense, including data collection methods such as Particle Image Velocimetry (PIV). Given that the scope of ESSD excludes interpretation of the data, we constrained ourselves to citing literature that illustrates the potential applications and usefulness of the data (for example, Le Boursicaud et al., 2016). While PIV analysis of the data is currently underway and could technically be included in the manuscript, we understand on the basis of the guidelines that this would distract from the data that is being presented and transcend the scope of the paper.

| | |
|---|---|
| **RC2.b** | *Second, the authors are honest, revealing the limits and uncertainties affecting their database. They often go too much into the details in describing these uncertainties or errors. Some described errors are basic - time shifts between sensors, sensitivity of the measurements to temperature and even perturbations due to external elements (such as pedestrians perturbing radar measurements). Some of these errors should be avoided if possible during the experiments, other should be filtered out during data processing. The critical analysis of data sets left to the users should be limited. The users should be able to download what can be considered as high quality data. Some sentences are worrying in the text : "The offsets were only determined for the last two days (, since the experiments conducted previously were not judged to be of sufficient quality...", suggesting that the data set contains data of highly variable quality and that the authors let the users make their own selection. This is not really acceptable and compatible with the quality requirements of the data sets presented in ESS.* |

**Authors**: In the original manuscript, the data was documented in a way that users who desire to use the full raw data were informed of all known limitations, even the issues that were amended by preprocessing. Thanks to the reviewers' comments, we have realized that the goal of ESSD publications is to present high-quality data rather than the detailed data collection process. We have reorganized and reprocessed the data, and will publish two high-quality versions of the data, each having a designated quality standard and area of application: flood monitoring

> research and flood model calibration research. In this spirit, the elementary and overly technical descriptions of challenges have been removed from the manuscript to the supplementary material. Information that is not relevant to the two high-quality data sets has been removed from the paper.

**RC2.c**   *Third, the experimental setting is relatively impressive (volume, surface) but represents an extremely simple system : one single manhole. I have some doubts that such a simple experiment provides original and new data sets to test and calibrate urban flood simulation models. At least this should be illustrated in the manuscript.*

**Authors**: We recognize that the description of the experimental layout is not made clear in the manuscript, and we have therefore replaced Figure 1 with a 3D computer rendering of the facility (see reply to comment RC2.4). In the new figure, the features of the setup are more evident (sewer inlets, manholes, multiple pipes, open channels, and ponds). The experimental setup was designed so that principal phenomena characteristic of urban floods, such as manhole surcharging, street curb flow, and basement flooding, would be simulated despite the restricted facility size.

**RC2.3**   *Finally, the figures in the manuscript are hardly readable and should be significantly improved.*

**Authors**: The figures have been improved to increase readability and captions edited to be more descriptive.

**Replies to specific comments**

These replies address the comments made in the supplement uploaded by the reviewer. The margin notes on the left indicate where the reviewer comment was made in the original manuscript.

   **RC2.1**   *Is really the roughness a dominant factor? Topography and road network geometry are certainly factors of primary importance.*

**Authors:** The statement was made in the context of urban flood model calibration. According to the study referenced (Hunter et al, 2008), roughness is a dominant factor in flood models for which topography/building data is of sufficient quality (the study uses a 2 meter resolution DEM modified to include buildings and curbs). Also, with regards to model calibration, roughness is usually regarded as a parameter that has to be calibrated whereas topography/road network/ building footprints are used to create the model layout and are often assumed to be correct. Besides surface roughness, other model parameters will also be able to be calibrated, e.g. weir and manhole discharge coefficients, pipe roughness.

**Changes**: No changes were made.
* * *
<table>
<tr><td rowspan="2">Page 2
Line 18</td><td>RC2.2     This should be somehow illustrated in the manuscript.</td></tr>
<tr><td>Authors: We understand the concern of the reviewer but we have intentionally omitted such data analyses (currently ongoing) from the manuscript. According to the ESSD Aims and Scope statement, "Articles in the data section may pertain to the planning, instrumentation, and execution of experiments or collection of data. Any interpretation of data is outside the scope of regular articles." (http://www.earth-system-science-data.net/about/aims_and_scope.html). Given that the flood measurement concepts mentioned are non-standard transformations of the data, we judge them to be outside the scope of the article.
According to the ESSD review criteria, "It should be plausible that the data, alone or in combination with other data sets, can be used in future interpretations, for the comparison to model output or to verify other experiments or observations." (http://www.earth-system-science-data.net/peer_review/review_criteria.html). We argue that, on the basis of the literature cited (Le Boursicaud et al., 2016; Dramais et al., 2011) and the extensive documentation of the experiments, the collected images can plausibly be used for developing novel flood monitoring methods. The use of robust temperature sensors for flood monitoring is arguably less plausible, since there is no guarantee that (storm)water temperature and air temperature differ.</td></tr>
</table>

**Changes**: The temperature data collected at one of the manhole openings is not presented in the manuscript anymore as a novel data type for flood monitoring.

**Page 2**
**Line 24**

***RC2.3*** *The main questions are not related to the models but to their implementation. Data are also often available now as mentioned by the authors (pictures, films, post-event surveys), but hardly valuable in real-time.*

**Authors:** Indeed, the models and raw data are often available, but assimilation of the data to enhance model performance is still an issue. The developments that could be made possible (we cannot prove it in the manuscript, but it is very plausible) thanks to the floodX dataset are precisely methods for (i) automatically interpreting image/sensor data into information consistent with model variables and (ii) assimilating the data into the model to improve predictive power. These two goals have been now listed explicitly in the introduction and in a new section.

**Changes:** These two goals have been now added explicitly to the introduction of the manuscript:

In summary, the floodX data will be used for two distinct but related areas of urban flood research: (i) automatic interpretation of image data into useful information for flood monitoring and model validation, and (ii) development of flood model calibration methods with overland flow data to improve the predictive power of the models.

The potential applications of the data are now presented in the revised manuscript in a new section (section 7):

Monitoring data for historic urban pluvial floods is typically limited to the (underground) drainage network because most sensors are designed specifically for that setting. Overland flow and accumulation is of much more interest than the drainage network when modelling urban pluvial floods, but the lack of suitable sensors means that flood hydrologists must often calibrate and validate their models with very limited or partial information. The data collected in the floodX project will be used to develop and investigate both image-based flood monitoring methods and flood model calibration schemes that can assimilate non-standard overland flow data. The tools developed in these two lines of research are necessary for using social media images and surveillance videos from real flood events to create more reliable flood models.

A trove of overland flow information lies in existing surveillance infrastructure and social media in the form of images and videos. With appropriate processing methods, flow and water depth could be automatically estimated from these data, and the floodX datasets[1] provide an opportunity to research such methods. For example, the measurement of shallow overland flows with Large Scale Particle Image Velocimetry could be investigated in channel c3 with the two cameras and two radar systems (Figure 1). For the moment, LSPIV has been investigated in urban settings only for large flows and without direct validation data (Guillén et al., 2017; Perks et al., 2016), or for seeded flows (Branisavljević and Prodanović, 2006). Another example is the use of deep learning to estimate flood water levels through semantic scene interpretation, e.g. by interpreting the immersion level of objects of known dimensions in snapshots and videos (Figure 2).
* * *
[1] floodX Preprocessed Monitoring Data, floodX Flooding Videos, floodX Flooding Images

[Figure]

**Figure 1: View from camera CAM2 in which channel c3 is visible, as well as the scaffolding holding two radar-based flow measurement systems. The same channel is also visible from camera CAM3.**

[Figure]

**Figure 2: View from camera CAM1 in which a bicycle is visible in the flood water behind the dam. Deep learning could make it possible to automatically estimate the water level from such an image. An ultrasonic sensor above the water and a pressure sensor in manhole m1 provide water level data.**

The floodX data can also be used for urban pluvial flood model calibration research since the flood monitoring setup was contained within a hydraulic system comparable to an urban catchment. Thanks to this unique setup and the subset of calibration-quality data[2], it will be possible to test model calibration concepts capable of assimilating the overland flooding data delivered by the novel monitoring methods. The specific questions one may need to address are, for example, the choice of appropriate objective functions and of an appropriate weighting strategy.

In summary, the research made possible by the floodX data will both contribute to urban flood monitoring innovation and improve the reliability of urban flood modelling, thereby increasing the effectiveness of urban flood management services, such as flood forecasting, response, and risk management.
* * *
[2] floodX Preprocessed Calibration Data

**Page 3**

**Line 6**

***RC2.4***     *The drainage system, manholes and constructions are not really visible on this picture.*

**Authors:** Following a similar comment from reviewer 3, the figure has been replaced with a computer rendering of the setup in which the drainage system, manholes, and constructions are visible, along with arrows indicating flow direction.

**Changes:** The following figure has replaced Figure 1:

[Figure]

Figure 1: Computer rendering of the floodable area of the flood facility, illustrating main hydraulic flows on the surface (blue arrows), and in pipes (red arrows). The labels indicate major hydraulic components and the placement of sensors, with the IPC labels indicating what component is in the centre of each camera view.

**Page 3**
**Line 14**

***RC2.5***     *Not very clear. Explain what has been difficult.*

**Authors:** The manuscript now explains the cause of the measurement challenges.

**Changes:** The following sentence has been adapted (section 2):

> The measurement of flow at the inlet and outlet pipes of the system (p1 and p6, **Error! Reference source not found.**) was a challenge because of the non-laminar flow conditions caused by a turn in the pipe or the presence of valve v4, respectively. For this reason, the flow meters were installed in specially designed pipe extensions as documented in the supplementary material.

**Page 4**
**Line 4**

***RC2.6***     *Not very clear, a little too schematic. It seems that the model includes one single pipe and one single manhole according to this figure which is very simplified if compared to "real" urban flood situations.*

**Authors:** Figure 2 has been adapted to make it easier to read. In combination with Figure 1, where the same style has been used, the readers will now be able to visualize the experimental setup.

**Changes:** Figure 2 has been adapted in the following manner: the same symbols, colors, and line styles are used as in Figure 1; manhole openings and orifices are indicated differently to make them more visible, overland flow channels are included.

[Figure]

Figure 2: Schematic representation of hydraulic network, including the labels of the main hydraulic components and the locations of various sensor types.

| Page 4 Line 17 | **RC2.7** | *The 2-dimensional (xy) features of urban floods is not represented here.* |

**Authors:** Two-dimensional information of the experimental setup can be found in the floodX documentation package (layout of facility, elevation data).
Regarding information about 2D flow, one of the cameras (CAM4) is directed at channel c4. This channel is sufficiently wide so that flow can spread out in two dimensions (contrarily to other channels for which flow is constrained in one direction). Thanks to the radar systems installed in channel c3 right before channel c4, the flow in channel c4 can be known precisely.

**Changes:** No changes were made to the manuscript.

| Page 5 Line 6 | **RC2.8** | *I am wondering if this description is really useful…* |

**Authors:** The nomenclature description has been removed from the manuscript (similar point raised by Reviewer 3). It is available in the supplementary material provided with the paper and in the floodX Documentation package.

**Changes:** The nomenclature of data sources has been moved to the supplementary material.

| Page 5 Line 19 | **RC2.9** | *already described* |

**Authors:** Thank you for pointing this out.

**Changes:** The redundancy has been amended.

| Page 5 line 23 | **RC2.10** | *if the valve v2 is closed than the simulated system is reduced to a direct overflow over a plane area combined with a single manhole overflow… What can we really learn about urban floodings based on this very simplified experiment.* |

**Authors:** The system is not limited to one pipe, one manhole, and one surface runoff but includes multiple pipes and surface flows, as well as ponding, three weirs, and basement flooding. The experiment is more clearly presented in the revised manuscript.

**Changes:** The presentation of the experimental setup has been made clearer in the manuscript. Specifically, we have improved Figure 1 and Figure 2 as described above. Minor changes were made to the description of the hydraulic network in section 2 as well.

| Page 5 Line 31 | **RC2.11** | *this command refers to the program used to handle the measured data. Impossible to understand for the readers.* |

**Authors:** The authors acknowledge the reviewer's comment. Also reacting to similar comments from reviewers 2 and 3, the data organization has been updated to make reuse simpler: "floodX dataset" has been divided into three datasets: 1) raw data and processing code, 2) clean, highquality data to be used for model *calibration* research, 3) clean data, high-quality data to be used for new flood *monitoring* methods research. Indeed, the typical user is not expected to have to process the code themselves.

**Changes:** The sentence has been removed. Data is now organized in such a way that users of the data do not need to come into contact with the processing code.
* * *
**Page 6
Line 3**

**RC2.12**   *Not if the inflow is automatically controlled.*

**Authors:** Inflow could unfortunately not be controlled automatically.

**Changes:** Slight modification to the following sentence (section 4.2, page 5):

- The valve controlling the inflow to the system had to be controlled manually and therefore the flow commands had to be simply defined.
* * *
**Page 6
Line 9**

**RC2.13**   *how where the volumes and discharges selected ? Empirically or based on the capacities of the components of the system ?*

**Authors:** The volumes and discharges were chosen based on the dynamics of the system, i.e. in order to produce flood phenomena of interest: manhole overflow, pond spillage, basement flooding. Empirical tests were carried to determine what hydrographs produced which phenomena.

**Changes:** The following sentence was rephrased to make the design criteria and procedure clearer:

The hydrographs of the flood events were defined through trial and error to produce a range of system responses (e.g., occurrence of dam overflow, flooding in basement).
* * *
**Page 6
Line 10**

**RC2.14**   *Does this has any effect on the response of the system ? Evaporation is certainly limited.*

**Authors:** This does not affect the response of the system, but could possibly have an important effect on image-based flood measurement techniques (e.g. PIV). This question is one that is being addressed with ongoing research with the floodX data.

**Changes:** No changes made.
* * *
**Page 6
Line 13**

**RC2.15**   *I do not understand what the authors mean here.*

**Authors:** This comment is linked to the comment at line 9 regarding the definition of flood hydrographs.

**Changes:** Refer to reply to comment RC2.13 (page 6, line 9).

<table>
<tr><td>Page 6
Line 19</td><td>RC2.16</td><td>Experiments considered as not satisfactory could also as well have been removed from the data set.</td></tr>
</table>

**Authors:** Based on this comment and comments from reviewer 3, we now define two data quality standards: *calibration quality* and *monitoring quality.* Experiments that do not meet these standards are tagged as such and removed from the final data sets.

**Changes:** In Section 4.3 (Pages 5 and 6 in the revised version of the manuscript), the text regarding data quality has been revised as follows:

The experiments conducted produced data with varying degrees of quality. Three levels of experiment data quality have been defined in order to facilitate reuse of the data:

- **Insufficient quality** experiments have issues with sensor performance and experimental layout. These experiments are omitted from the preprocessed data and from the manuscript. However, they are still available in the raw data and documented in the metadata.
- **Monitoring quality** experiments provide data that can be used for investigating novel flood monitoring approaches. The data provided is correct but the hydraulic system (e.g. valve opening) is not necessarily the same between experiments.
- **Calibration quality** experiments are a subset of the monitoring quality experiments and can be used to calibrate and validate urban flooding and drainage models. The hydraulic system is the same for each experiment, only the hydrograph differs.

Each experiment has been tagged according to its level of quality in the experiment metadata file, which can be found in the *floodX Raw Data, Metadata, and Preprocessing Code* package (doi: 10.5281/zenodo.830505). Datasets for both experiment subsets (monitoring or calibration) have been exported to distinct data packages[3].

<table>
<tr><td>Page 6
Line 25</td><td>RC2.17</td><td>Did the authors not try to analyse the videos (PIV) ? If not, why and what is the usefulness of these videos.</td></tr>
</table>

**Authors:** Please refer to the answer to comment RC2.2 (page 2, line 18).

<table>
<tr><td>Page 6
Line 30</td><td>RC2.18</td><td>A very technical comment. The readers are not necessary interrested in the technical problems or limitations. What is important is the final dataset.</td></tr>
</table>

**Authors:** See reply to following comment RC2.19 (page 7 line 12).
* * *
[3] *floodX Preprocessed Monitoring Data* (doi: 10.5281/zenodo.830511) and *floodX Preprocessed Calibration Data* (doi: 10.5281/zenodo.830513)

**Page 7**
**Line 12**

> **RC2.19**   *Far too technical description, refering to methods (OCR…) that are not familiar for most readers. The authors must either remain simple summarising the principles or give some more details and illustrations of the methods.*

**Authors:** We now understand that the description is too technical for the typical reader.

**Changes:** The description OCR has been removed from the manuscript and moved to the supplementary material. OCR is now only briefly mentioned in the manuscript (section 4.4):

> In order to reach higher sampling rates for certain sensors, images and videos of the sensor displays were made and analysed with optical character recognition (OCR)[4].

**Page 7**
**Line 17**

> **RC2.20**   *Useless level of detail.*

**Authors:** The authors agree with the reviewer.

**Changes:** The level of detail has been reduced to a minimum and the information has been moved to the readme of the metadata files.

> The following transformations were applied to preprocess the data: consolidation of multiple data files, chronological sorting, reformatting of date, time, and null values, correction of temporal offsets, removal of extreme and impossible values, and segmentation of data into a separate file for each experiment.

**Page 7**
**Line 24**

> **RC2.21**   *All these treatments are common treatments to provide high quality data sets. It is useless to describe them with such details.*

**Authors:** See answer to comment RC2.20 (page 7, line 17).

**Page 7**
**Line36**

> **RC2.22**   *Strange recommendation of data viewing and processing software....*

**Authors:** The authors agree with the reviewer.

**Changes:** The referral to the (open source) software has been removed.
* * *
[4] *floodX Data Logger Images* (doi: 10.5281/zenodo.830507) and *floodX Data Logger Videos* (doi: 10.5281/zenodo.830504)

**Page 8**
**Line 2**

*RC2.23    The legends are much too small. Please increase. These graphics can be of little use for the readers. Moreover, these graphics should be commented to illustrate the usefulness of the data set.*

**Authors:** The authors agree with the reviewer that the utility of this figure is questionable.

**Changes:** This figure has been removed.

**Page 8**
**Line 10**

*RC2.24    The aim of ess is to present good-quality dataset. The questions related to data filtering and correction should be described and solved.*

**Authors:** The data has been cleaned and/or documented to a level that makes it usable by anyone interested in using the data sets.  The further improvements suggested in the first version of the manuscript are noncompulsory and therefore were removed.

**Changes:** The section has been revised to remove the suggestions.

**Page 8**
**Line 14**

*RC2.25    This is a very standard question that the authors have to solve and not share with the readers and the future users of their database.*

**Authors:** The information about temporal offsets has been moved to the documentation package and removed from the manuscript. The information is also directly included with the video data.

**Changes:** The information about temporal offsets has been moved to the documentation package and removed from the manuscript.

**Page 9**
**Line 7**

*RC2.26    Again, this is a very trivial problem that has to be solved during the experiment.*

**Authors:** The authors agree with the reviewer. See reply to the comment RC2.25 (page 8, line 14).

**Page 9**
**Line 9**

*RC2.27    If the data is considered of bad quality, it should not be provided in the dataset !!!!*

**Authors:** With the differentiated quality levels introduced in the revised version of the manuscript (see reply to comment RC2.16 (page 6, line 19), it now makes sense to determine this information for previous experiments as well. The offsets have been estimated for remaining experiments and provided with flooding videos package.

**Changes:** The sentence has been removed since the offsets are now determined for all videos.

**Page 9**
**Line 20**

*RC2.28    If it has been moved, its name could have been changed...*

**Authors:** The reviewer's suggestion is justified but the initially submitted version uses precisely the same logic: at the new location, the sensor received a different name. In this way the data from the

two locations is completely distinct. However, since data from this sensor is no longer included in the final data sets, this information is no longer required in the manuscript.

**Changes:** The sentence has been removed.

**Page 9**
**Line 27**

**RC2.29**  *Could this be verified rather than only hypothesized*

**Authors:** This hypothesis is made on the basis of the sensor user manual which states that the ultrasonic distance measurement is corrected with an internal temperature sensor. Since the measurement is local, the correction can be erroneous if there is a temperature difference over the air gap of the measurement.

**Changes:** The background information for this hypothesis has been added to the script in section 5.2.1:

During periods in which the dam is empty, a drift of up to 7 mm in the sensor measurement can be noticed (**Error! Reference source not found.**). These ultrasonic sensors use the speed of sound, which is temperature-dependent, to estimate distance. The drift is possibly linked to direct solar radiation on the sensor body, which would raise the internal temperature of the sensor and cause it to overestimate the ambient air temperature for which there is a compensation.

**Page 10**
**Line 3**

**RC2.30**  *I do not understand what we should look at. What does this figure represent (successive filling and emptying ?). Where is the drift ? What do the red circles show ?*

**Authors:** We agree that the figure was not clear.

**Changes:** The legend of the figure has been made more descriptive and the clarity of the figure has been improved.

[Figure]

Figure 3: Water level behind dam s3 measured by ultrasonic sensor s3_h_us_maxbotix. The measured level in absence of water can be seen varying over time. At 14:30 (rectangle to the right), the level is lower than at 13:00 (rectangle to the left).

**Page 10**
**Line 7**

**RC2.31**   *Also unclear...*

**Authors:** The data logger did not allow negative water values, so even if drift was present in the measurement it would not always be visible in the data. We considered that this information was too technical for the purpose of the paper.

**Changes:** The sentence has been removed from the manuscript.

**Page 10**
**Line 11**

**RC2.32**   *Why did people walk in front of the radar. Were the experiments not conducted without interventions ?*

**Authors:** People did not walk in front of the radar but near it. If no flow was present, then even nearby movement could be interpreted as flow. This highlights a weakness of radar for measuring flow over urban surfaces. The experiments were conducted without interventions to the setup, but movement within the flood facility was allowed to monitor data collection and experiment progress.

**Changes:** The description of the data anomaly is described more clearly now. Additionally, these outliers are now removed from the final datasets during preprocessing. The sentences now read:

In the absence of flow and when a person walked near the measurement location, this sensor produced flow estimations. These false measurements have been removed from the datasets. When flow is present, the passage of people on the crossing does not appear to effect the measurement.

**Page 11**
**Line 6**

**RC2.33**   *Again, the authors should not share all their implementation difficulties with the reader (it is not the objective of ees) but provide good quality and useful datasets.*

**Authors:** Similar suggestions were made by reviewer 3, so the manuscript was amended to focus less on technicalities and more on the published data.

**Changes:** The data from the radars have been cleaned to remove the disturbances caused by people passing by (see reply to previous comment). The paragraph referred to by the reviewer is understandably not fitting to the purpose of the manuscript and has been removed.

**Page 11**
**Line 12**

**RC2.34**   *Useless details.*

**Authors:** The authors agree with the reviewer.

**Changes:** The useless details have been removed from the manuscript.

**Page 11
Line 15**

**RC2.35**  *That is really problematic ! Any explanation for this leakage ?*

**Authors:** This problem was also highlighted by reviewer 3. We concluded that the main reason for discharge differences between pipes p1 and p6 had to be coming from the discharge measurement in pipe p6. The possibility of leaks is not plausible because the volumes involved are too large. The discharge measurement in pipe p1 is judged accurate because it is confirmed by the measurement in pipe p3 and because the measurement conditions in pipes p1 and p3 are much better than in pipe p6. The measurement in pipe p6 was highly constrained because of the facility layout, and despite having installed additional piping, the stabilization distance before and after the measurement was not ideal. Reinforcing this conclusion is the discovery of blatant measurement errors in the discharge data of pipe p6.

**Changes:** *The discharge data measured in pipe p6 has been removed from the preprocessed datasets but was left in the raw data. Because of this decision, the description of the issue has been shortened and mentioned in the context of "Data omitted from the datasets":*

Over all experiments, the discharge measured by the ultrasonic discharge sensor p6_q_us_nivus at the facility outlet deviates substantially from the discharge measured at the inlet of the flood facility in pipe p1. Different hypotheses to explain the volume differences were brought forward, such as residual water in the system and pipe leaks, but investigation of the data and the facility plans led to the conclusion that such factors could not fully explain the discrepancy. The only remaining explanation is that the constrained measurement conditions, especially the short stabilization distance before and after the sensor, the presence of a valve at the end of the pipe, and the frequent changes between full and partially filled pipe, caused the measurement to be erroneous. This conclusion is corroborated by the discovery of artefacts in the discharge data. Since the volume differences are larger than the expected measurement error for the technology (DWA, 2011), the data was judged to be of insufficient quality and, therefore, was removed from the datasets.

**Page 11
Line 26**

**RC2.36**  *This does not explain losses*

**Authors:** This does not explain overall water loss (measured over multiple experiments), but it does explain water loss or gain between individual experiments.

**Changes:** No action was taken

**Page 12
Line 9**

**RC2.37**  *The lack of explanation and of control of a basic aspect of the experiment (water balance) is not satisfactory at all.*

**Authors:** The authors agree with the reviewer's comment. Please refer to the reply to comment RC2.35 (page 11, line 15).

**Page 12**
**Line 19**

***RC2.38***  *The experiment does not appear complex at all...*

**Authors:** We experienced the experiment as complex because of the diversity of sensors, the size of the system, and short experimentation time. However, perhaps the experiment would have been straightforward for more experienced researchers.

**Changes:** We have removed the qualifier.

**Page 14**
**Line 13**

***RC2.39***  *It is nevertheless not shown how the measurements of the sensor can be combined to better measure and understand an urban flood. This is the main challenge.*

**Authors:** Please refer to the answer of RC2.2 (page 2, line 18).

**Page 14**
**Line 14**

***RC2.40***  *This is not demonstrated at all.*

**Authors:** Please refer to the answer to RC2.2 (page 2, line 18). According to ESSD review criteria, "It should be plausible that the data, alone or in combination with other data sets, can be used in future interpretations, for the comparison to model output or to verify other experiments or observations." ([http://www.earth-system-science-data.net/peer_review/review_criteria.html](http://www.earth-system-science-data.net/peer_review/review_criteria.html)) Based on these directives, we understand that the demonstration of data's potential lies in the realm of data interpretation and is outside the scope of ESSD data articles , and are therefore not included in this manuscript.

**Page 14**
**Line 16**

***RC2.41***  *The authors should have tested this aspect before submitting a manuscript.*

**Authors:** See reply to previous comment RC2.40 (page 14, line 14).

**Page 14**
**Line 17**

***RC2.42***  *This is absolutely not illustrated...*

**Authors:** We agree with the reviewer's comment since the manuscript did not discuss temperature-based flooding detection in the text.

**Changes:** The manuscript has been edited to remove this application of the data, since it is not completely plausible that it is possible.

**Page 14**
**Line 19**

***RC2.43***  *The authors should also give an example of how their dataset can help calibrating urban flood models.*

**Authors:** Please refer to the answer to RC2.2 (page 2, line 18).

**References**

Le Boursicaud, R., Pénard, L., Hauet, A., Thollet, F. and Le Coz, J.: Gauging extreme floods on YouTube: application of LSPIV to home movies for the post-event determination of stream discharges, Hydrol. Process., 30(1), 90–105, doi:10.1002/hyp.10532, 2016.

Branisavljević, N. and Prodanović, D.: Large Scale Particle Image Velocimetry – Measuring Urban Discharge, Vodoprivreda, 38(4–6), 233–238 [online] Available from: http://www.vodoprivreda.net/large-scale-particle-image-velocimetry-merenje-urbanog-oticaja/ (Accessed 18 March 2016), 2006.

Dramais, G., Le Coz, J., Camenen, B. and Hauet, A.: Advantages of a mobile LSPIV method for measuring flood discharges and improving stage–discharge curves, J. Hydro-environment Res., 5(4), 301–312, doi:10.1016/j.jher.2010.12.005, 2011.

DWA: Merkblatt DWA-M 181 Messung von Wasserstand und Durchfluss in Entwässerungssystemen, DWA Deutsche Vereinigung für Wasserwirtschaft, Abwasser und Abfall e. V., 2011.

Guillén, N. F., Patalano, A., García, C. M. and Bertoni, J. C.: Use of LSPIV in assessing urban flash flood vulnerability, Nat. Hazards, doi:10.1007/s11069-017-2768-8, 2017.

Perks, M. T., Russell, A. J. and Large, A. R. G.: Technical Note: Advances in flash flood monitoring using unmanned aerial vehicles (UAVs), Hydrol. Earth Syst. Sci., 20(10), 4005–4015, doi:10.5194/hess-20-4005-2016, 2016.

---

## Author Comment (AC3) · 19 Jul 2017

**AUTHORS' REPLIES TO RC3 SUPPLEMENT**

**General comments**

| | |
|---|---|
| **RC3** | *The paper describes experimental data sets of urban flooding experiments. The experimental set up covers a 500 m² area which provides data at almost a 1:1 scale. Such data sets are not very common and can be of high interest to test urban flood models in conditions close to field conditions. The authors should emphasize this point in their description of the data set. I agree with the authors that such a data set is very valuable to assess the validity of urban flood models.*

*If we refer to the scope of ESSD, it is stated that "Any interpretation of the data is outside the scope of a regular paper….. Any comparison to other methods is beyond the scope of regular data." Therefore, it is not expected to see comparisons of the data with numerical models in the paper, nor any complete description or interpretation of the data. However, ESSD expects good quality data sets and the description should be clear and complete enough to provide the relevant information useful for potential users. I believe the authors did not really succeed to reach this last goal for the following reasons:* |
| **Authors**: | The authors value the helpful suggestions provided by the reviewer. Thanks to the general and detailed feedback, the manuscript has been modified to provide a clearer understanding of the urban flooding data we are sharing. |

| | |
|---|---|
| **RC3.a** | *the experimental set up description provided in the paper is not clear and complete enough to really understand the experimental set up and the various components of the experiment. The photo provided in Figure 1 does not provide the location of the sensors and does not allow understanding how the experiment was conducted and how water was flowing. Figure 2 provides a schematic transverse view of the experiment but it gives the impression that it is only a connection of pipes and valves, and does not inform on the lateral extension of the experiment. My point of view is that the authors should provide one or more plan view of their experiment showing the interconnections between the various elements, the 2D extension of their experiment and of water flow and also the locations of the various sensors (and possibly the view angles of the cameras). The authors make reference to the floodX documentation data set that provides plans of the experimental set up, but I believe this information should be provided in the main paper, in a schematic manner, but providing information on the main elements of the experiments (reservoirs, pipes, valves, manholes, etc..), of the location of the sensors and of the main directions of the flow.* |
| **Authors**: | *We agree with the reviewer that Figure 1 did not serve its purpose of giving a good overview of the flood facility. A computer rendering of the facility was created to replace Figure 1. With such a rendering, the complete floodable area is visible.* |
| **Changes**: | Figure 1 has been replaced with a 3D computer rendering of the flood facility supplemented with labels indicating the most important hydraulic components and sensors. The same |

symbols as for Figure 2 have been used, making the two figures easier to interpret together.

[Figure]

Figure 1: Overview of the floodable area of the floodX flood facility, illustrating main hydraulic flows on the surface (blue arrows), and in pipes (red arrows). The labels indicate major hydraulic components and the placement of sensors, with the IPC labels indicating what component is in the centre of each camera view.

| RC3.b | in the abstract the authors mention 37 experiments, but the paper does not provide an overview of the content of those experiments. The reader should refer to the data set documentation for that. I believe that a summary table, providing the main features of the experiments (duration, input discharge, specific configurations, etc.) should be provided in the paper, together with information on the data reliability of each data set (see next point). The potential user of the data should have guidelines to determine which experiment is reliable and which one is relevant for his/her specific needs. |

**Authors**: We agree with the reviewer that such a table is lacking from the manuscript, so such a table was included in the revised version of the manuscript. Based on the feedback from reviewers 2 and 3, we have defined two high-quality standards for experiments, one being suitable for flood monitoring research and the other being suitable for flood model calibration research as well as the flood monitoring; this information was also included in the table. Additionally, since only 21 of the experiments pass the newly defined quality standard criteria, the abstract has been adapted to reflect these changes.

**Changes**: A table has been created and added to the section in which the experiments are described.

Table 3: Selection of high-quality experiments conducted, including the duration of the flooding, the total volume of water introduced in the system, and the experiment quality. The experiments are sorted by their total flood volume.

| # | Duration | Volume [m³] | Description | Lighting | Experiment quality |
|---|----------|-------------|-------------|----------|--------------------|
| 29 | 00:09:50 | 3.5 | small event | night + infrared | calibration |
| 14 | 00:07:00 | 3.8 | small event | overcast | monitoring |
| 35 | 00:15:00 | 3.8 | small, empty sewers, manhole cover on m2 | overcast | monitoring |
| 16 | 00:05:00 | 4.0 | small event, full sewers at start | direct sun | monitoring |
| 30 | 00:05:15 | 4.2 | small event | night + flood light | calibration |
| 26 | 00:07:00 | 4.8 | small event, full sewers at start | evening sun | calibration |
| 17 | 00:06:00 | 5.0 | two small events, full sewers at start | direct sun | monitoring |
| 25 | 00:06:00 | 5.3 | small event | evening sun | calibration |
| 27 | 00:07:00 | 6.6 | two small events | evening sun | calibration |
| 15 | 00:07:00 | 7.0 | two small events | overcast; direct sun | monitoring |
| 34 | 00:20:00 | 12.8 | multiple small, slow events | overcast | calibration |
| 31 | 00:16:00 | 24.2 | simple dam overflow with pre-impulse | night + flood light | calibration |
| 18 | 00:15:00 | 25.3 | simple dam overflow | direct sun | monitoring |
| 21 | 00:17:00 | 26.5 | simple dam overflow with pre-impulse | direct sun | monitoring |
| 28 | 00:23:00 | 27.6 | simple dam overflow | evening sun | calibration |
| 23 | 00:19:00 | 31.4 | slow event | overcast; direct sun | monitoring |
| 33 | 00:20:40 | 32.2 | large event followed by a small one | overcast | calibration |
| 20 | 00:16:00 | 33.3 | multiple large events | overcast; direct sun | monitoring |
| 22 | 00:20:40 | 34.5 | large event followed by a small one | direct sun | monitoring |
| 19 | 00:15:00 | 44.9 | multiple large events | direct sun | monitoring |
| 32 | 00:22:00 | 64.1 | multiple large events | night + infrared | calibration |

**RC3.c**    *the authors also provide the codes used to pre-process the data. They provide the raw and pre-processed data which I find a good point, as the potential users have the possibility to take the raw data and make their own pre-processing if they do not agree with the authors' one. However, the pre-processing data presentation is somehow puzzling for the reader. As the authors are honest and do not hide the problems encountered with their data, the reader is left with the idea that none of the data set is of enough quality. I think the authors should provide clearer information on the data set quality, and if some experiments are not reliable enough, they should consider removing them from the data set.*

**Authors**: We agree with the reviewers (reviewer 2 had a similar comment).

**Changes**: First, the table added on the basis of comment RC3.b includes information about the quality of the data and its recommended use, be it for flood model calibration or flood monitoring. Experiments of insufficient quality have been removed from the dataset. Second, we have reviewed the manuscript and removed parts that give unnecessary weight to standard data issues such as temporal shift that we were able to resolve. Third, we have changed the dataset structure to include two distinct data packages, each containing a cleaned selection of data of different quality standards, the first being appropriate for flood monitoring research and the second being appropriate for model calibration research. Finally, we have reduced the number of experiments we "advertise" in the abstract to reflect the number of high-quality experiments only.

| | |
|---|---|
| ***RC3.d*** | *the authors explain with quite details, the problems they encountered with their experiments. This is a good point for the future users. However, the presentation is not balanced enough and the reader ends its reading with serious doubts about the interest of the data set. I believe the authors should also spend some times explaining what makes their data set valuable for other users, and what the strengths of their data sets are. Figure 3 is a nice summary of the collected data, but should be commented more in details to explain what can be seen in the figure. In particular the use of the temperature sensors is presented as new, but this would gain being illustrated with some examples. I also believe the OCR treatment of the data loggers is an original application that contributes to the originality of the data set.* |

**Authors**: This comment links to the previous one (RC3.c) with regards to the unbalanced presentation of the weaknesses and potential of the data. We acknowledge the comment and have added a section before the conclusion highlighting possible uses of the data, supported by recent literature.

**Changes**: The following section has been added to the manuscript:

**7. POTENTIAL APPLICATIONS OF DATASETS**

Monitoring data for historic urban pluvial floods is typically limited to the (underground) drainage network because most sensors are designed specifically for that setting. Overland flow and accumulation is of much more interest than the drainage network when modelling urban pluvial floods, but the lack of suitable sensors means that flood hydrologists must often calibrate and validate their models with very limited or partial information. The data collected in the floodX project will be used to develop and investigate both image-based flood monitoring methods and flood model calibration schemes that can assimilate non-standard overland flow data. The tools developed in these two lines of research are necessary for using social media images and surveillance videos from real flood events to create more reliable flood models.

A trove of overland flow information lies in existing surveillance infrastructure and social media in the form of images and videos. With appropriate processing methods, flow and water depth could be automatically estimated from these data, and the floodX datasets[1] provide an opportunity to research such methods. For example, the measurement of shallow overland flows with Large Scale Particle Image Velocimetry could be investigated in channel c3 with the two cameras and two radar systems (Figure 1). For the moment, LSPIV has been investigated in urban settings only for large flows and without direct validation data (Guillén et al., 2017; Perks et al., 2016), or for seeded flows (Branisavljević and Prodanović, 2006). Another example is the use of deep learning to estimate flood water levels through semantic scene interpretation, e.g. by interpreting the immersion level of objects of known dimensions in snapshots and videos (Figure 2).
* * *
[1] floodX Preprocessed Monitoring Data, floodX Flooding Videos, floodX Flooding Images

[Figure]

**Figure 1: View from camera CAM2 in which channel c3 is visible, as well as the scaffolding holding two radar-based flow measurement systems. The same channel is also visible from camera CAM3.**

[Figure]

**Figure 2: View from camera CAM1 in which a bicycle is visible in the flood water behind the dam. Deep learning could make it possible to automatically estimate the water level from such an image. An ultrasonic sensor above the water and a pressure sensor in manhole m1 provide water level data.**

The floodX data can also be used for urban pluvial flood model calibration research since the flood monitoring setup was contained within a hydraulic system comparable to an urban catchment. Thanks to this unique setup and the subset of calibration-quality data[2], it will be possible to test model calibration concepts capable of assimilating the overland flooding data delivered by the novel monitoring methods. The specific questions one may need to address are, for example, the choice of appropriate objective functions and of an appropriate weighting strategy.

In summary, the research made possible by the floodX data will both contribute to urban flood monitoring innovation and improve the reliability of urban flood modelling, thereby increasing the effectiveness of urban flood management services, such as flood forecasting, response, and risk management.

| | |
|---|---|
| ***RC3.e*** | *the authors should try to get a final version of the paper only focused on the data set description. So they may provide additional information (such as file naming, sensors drift, ..) in a supplementary material.* |

| | |
|---|---|
| **Authors**: | This point was also brought up several times by reviewer 2 and we have made changes so that the paper focuses more on the data and less on technicalities such as experimental challenges and data processing. |
* * *
[2] floodX Preprocessed Calibration Data

**Changes**: Many technical details have been moved out of the manuscript and into the supplementary material. Concerned passages include data logging with OCR (section 4.4.1), preprocessing of time series (section 4.4.2), preprocessing code (section 4.4.3). The discussion of temporal shift is more to-the-point (section 5.1), and we have removed anecdotal information about sensors that is not relevant for the data (section 5.2.2).
* * *
**RC3.f**     *at the very end of the paper, the authors mention water balance mass errors of about 20%. This is a big issue that may compromise the usefulness of the data set, and could require further analysis (in particular when putting doubts on the validity of the p6 discharge measurements). Shouldn't it be possible to make additional verifications to see if the proposed explanation may hold?*

**Authors**: The comment is valid and we have analyzed the data and the facility plans again to scrutinize the proposed explanations. We concluded that the main reason for discharge differences between pipes p1 and p6 had to be coming from the discharge measurement errors in pipe p6. The possibility of leaks is omitted because the volumes involved are too large. The discharge measurement in pipe p1 is judged accurate because it is confirmed by the measurement in pipe p3 and because the measurement conditions in pipes p1 and p3 are much better than in pipe p6. The measurement in pipe p6 was highly constrained because of the facility layout, and despite having installed additional piping, the stabilization distance before and after the measurement was not ideal. Reinforcing this conclusion is the discovery of blatant measurement errors in the discharge data of pipe p6.

**Changes**: The discharge data measured in pipe p6 has been removed from the preprocessed datasets but was left in the raw data. Because of this decision, the description of the issue has been shortened and mentioned in the context of section 5.3 "Data omitted from the datasets":

Over all experiments, the discharge measured by the ultrasonic discharge sensor p6_q_us_nivus at the facility outlet deviates substantially from the discharge measured at the inlet of the flood facility in pipe p1. Different hypotheses to explain the volume differences were brought forward, such as residual water in the system and pipe leaks, but investigation of the data and the facility plans led to the conclusion that such factors could not fully explain the discrepancy. The only remaining explanation is that the constrained measurement conditions, especially the short stabilization distance before and after the sensor, the presence of a valve at the end of the pipe, and the frequent changes between full and partially filled pipe, caused the measurement to be erroneous. This conclusion is corroborated by the discovery of artefacts in the discharge data. Since the volume differences are larger than the expected measurement error for the technology (DWA, 2011), the data was judged to be of insufficient quality and removed from the datasets.

| | |
|---|---|
| **RC3.g** | *In conclusion, I believe that, provided the authors better document the quality of the various data sets and better describe the experiments, their experiments provide valuable data sets at a 1:1 scale that are of great interest for the science community, in particular for evaluating urban flood models. The data set has also some potential to develop LS-PIV technique. However, in its present state, the data set presentation suffers too many weaknesses for the paper to be published. I recommend major revision of the paper, following the suggestions provided above and below, before possible publication in ESSD.* |
| **Authors**: | We would like to thank the reviewers for the time and effort invested in the review of the manuscript. The quality and value of the manuscript and the data has been greatly improved based on the reviewer's comments. |

**Specific comments**

| | |
|---|---|
| **RC3.1** | *Section 1.1. The authors may also consider the following references (Bazin et al., 2014; Mignot et al., 2013)* |
| **Authors**: | The publications are very interesting but unfortunately do not fit in line with the introduction in its current form. |
| **Changes**: | No changes were made. |

| | |
|---|---|
| **RC3.2** | *P. 2, line 12 "that covers the majority of the components of the hydraulic system": be more specific, which components?* |
| **Authors**: | The statement was imprecise and has been changed. Figures 1 and 2 have been updated to give an idea of the sensors' distribution in the hydraulic network. (See next comment RC3.3) |
| **Changes**: | The sentence has been changed:
"Second, the experiment has a relatively high density of sensors providing information on the major storage nodes and flow channels (Figure 1) so a comprehensive picture of the flooding dynamics can be gained." |

| | |
|---|---|
| **RC3.3** | *Figure 1, p.3 line 13: the figure is not clear enough and we do not see the elements mentioned by the authors. As mentioned before, one or several plan view of the experiment is necessary for the reader to understand how the experiment was working and the water flow paths. This is all the more necessary that Figure 2 gives the impression that the experiment is only a connection of reservoirs and pipes.* |
| **Authors**: | Figure 1 was indeed unhelpful and has been replaced with a 3D computer rendering of the facility and now includes the same labels as Figure 2. |
| **Changes**: | See reply to general comment RC3.a. |

| | |
|---|---|
| **RC3.4** | *P.5 put the section on files naming in a supplementary material, so that it is accessible to the reader without needing to download the whole data set.* |

**Authors**: This information was already included in the "floodX Documentation" package, but can also be provided as supplementary material to the paper.

**Changes**: This nomenclature information has been removed from the manuscript since it was also questioned by Reviewer 2. A document containing this information will be uploaded as supplementary material with the paper.

| | |
|---|---|
| **RC3.5** | *P. 6: section 4.3: I suggest that the authors provide a summary table with the various experiments that were conducted, their main characteristics and information about data quality.* |

**Authors**: See reply to general comment RC3.b.

| | |
|---|---|
| **RC3.6** | *Section 5.1: you could illustrate what the images look like by providing some examples.* |

**Authors**: The bulk of the OCR description has been moved to the supplementary material based on a comment from Reviewer 2, which falls in line with general comment RC3.e (that the paper should focus on the data and not experimental issues/data processing). An example image taken from one of the webcams is now provided in the supplementary material.

| | |
|---|---|
| **RC3.7** | *Figure 3 is a nice example of the collected data but it is hardly described and analyzed in the paper. The figure should be enlarged so that the authors can highlight the part of the curves that are providing useful information. Such a description would strengthen the paper, by providing an insight on how the data can be interpreted and used to the readers. In addition, the authors mention the novelty of the temperature sensors data, but they do not demonstrate it. An explanation/discussion of the corresponding figure would provide elements to really appreciate the interest of such data.* |

**Authors**: Figure 3 has been removed based on a comment from reviewer 2. However, as proposed by Reviewer 3, a section has been added discussing the utility and potential of the data.

**Changes**: see reply to general comment RC3.d.

| | |
|---|---|
| **RC3.8** | *P.9 line 19 " … was unknowingly moved.." . What does unknowingly means?* |

**Authors**: "Unknowingly" means it was moved without our knowing it at the time, but by looking at the videos and data we could determine a posteriori when and how it was moved. Data from this sensor are not included in the preprocessed datasets newly defined after the reviews, and so this sensor is not discussed anymore in the paper.

**Changes**: Mentions of this sensor have been removed.

| | |
|---|---|
| **RC3.9** | *Figures 4 and 5: the points the authors want to highlight are not very clear* |

**Authors**: This is a valid point also raised by reviewer 2.

**Changes:** The Figures have been made clearer and the legends made more descriptive.

[Figure]

**Figure 4: Measured water level in basement s6 by ultrasonic sensor s3_h_us_maxbotix presents abnormally low values after flooding (negative water levels of up to -7 mm can be seen in the red rectangle).**

[Figure]

**Figure 5: Signs of leakage visible in the water level at manhole m2 (sensor m2_h_p_endress_minilog). The pipe network should not be able to drain but over the course of two hours, starting at the red line, the water level in manhole m2 falls by around 1.7 meters.**

**RC3.10**   *Section 6.4: the authors should provide information on the experiments that were affected by this problem (in the summary table that would be added to the manuscript)*

**Authors**: Please refer to our answer to general comment RC3.f. The problem is due to unreliable discharge data from pipe p6 and this data has been removed from the preprocessed data.

**RC3.11**   *Section 6.6: among the difficulties mentioned by the authors, the water balance closure problem is certainly the most critical one. The authors propose several explaining factors, but none of them is fully convincing. In particular, problems with discharge measurements at the outlet are one plausible explanation. But could the authors make additional analysis to see if this explanation really holds?*

**Authors**: Please refer to our answer to general comment RC3.f.

**RC3.12**   *Figure 7: I don't understand what the "trend" line in the figure is.*

**Authors**: The trend is a logarithmic model fitted to the data. This figure has been removed from the paper since the discharge data from p6 is now discarded from the published datasets.

**RC3.13**   *P.13, line 1: I don't understand what the authors mean with this sentence.*

**Authors**: Thank you for pointing out the confusing sentence. We meant that for experiment setup and execution tasks that are novel in some way (in that experience is lacking with the method), three times more time should be planned for executing them.

**Changes:** This point is probably not very helpful since it is very subjective and for this reason has been removed.

**References**

Branisavljević, N. and Prodanović, D.: Large Scale Particle Image Velocimetry – Measuring Urban Discharge, Vodoprivreda, 38(4–6), 233–238 [online] Available from: http://www.vodoprivreda.net/large-scale-particle-image-velocimetry-merenje-urbanog-oticaja/ (Accessed 18 March 2016), 2006.

DWA: Merkblatt DWA-M 181 Messung von Wasserstand und Durchfluss in Entwässerungssystemen, DWA Deutsche Vereinigung für Wasserwirtschaft, Abwasser und Abfall e. V., 2011.

Guillén, N. F., Patalano, A., García, C. M. and Bertoni, J. C.: Use of LSPIV in assessing urban flash flood vulnerability, Nat. Hazards, doi:10.1007/s11069-017-2768-8, 2017.

Perks, M. T., Russell, A. J. and Large, A. R. G.: Technical Note: Advances in flash flood monitoring using unmanned aerial vehicles (UAVs), Hydrol. Earth Syst. Sci., 20(10), 4005–4015, doi:10.5194/hess-20-4005-2016, 2016.

---

## Author Comment (AC4) · 19 Jul 2017

Dear Eric Gaume,

We thank you for taking the time to provide your general and specific feedback on the manuscript. We have addressed the comments (including changes made to the manuscript) in our reply to Anonymous Referee #2, who we assume to be you due to the similarity of the comments. You will find our responses under the following link: https://www.earth-syst-sci-data-discuss.net/essd-2017-7/essd-2017-7-AC2-supplement.pdf

Kind regards,

Matthew Moy de Vitry (on behalf of the co-authors)